# VEGFR2 but not VEGFR3 governs integrity and remodeling of thyroid angiofollicular unit in normal state and during goitrogenesis

Jeon Yeob Jang[1,2,3,†], Sung Yong Choi[2,†], Intae Park[1,2], Do Young Park[1,2], Kibaek Choe[4], Pilhan Kim[4], Young Keum Kim[5], Byung-Joo Lee[3], Masanori Hirashima[6], Yoshiaki Kubota[7], Jeong-Won Park[8], Sheue-Yann Cheng[8], Andras Nagy[9], Young Joo Park[10], Kari Alitalo[11], Minho Shong[12,*] & Gou Young Koh[1,4,**]

## Abstract

Thyroid gland vasculature has a distinguishable characteristic of endothelial fenestrae, a critical component for proper molecular transport. However, the signaling pathway that critically governs the maintenance of thyroid vascular integrity, including endothelial fenestrae, is poorly understood. Here, we found profound and distinct expression of follicular epithelial VEGF-A and vascular VEGFR2 that were precisely regulated by circulating thyrotropin, while there were no meaningful expression of angiopoietin–Tie2 system in the thyroid gland. Our genetic depletion experiments revealed that VEGFR2, but not VEGFR3, is indispensable for maintenance of thyroid vascular integrity. Notably, blockade of VEGF-A or VEGFR2 not only abrogated vascular remodeling but also inhibited follicular hypertrophy, which led to the reduction of thyroid weights during goitrogenesis. Importantly, VEGFR2 blockade alone was sufficient to cause a reduction of endothelial fenestrae with decreases in thyrotropin-responsive genes in goitrogen-fed thyroids. Collectively, these findings establish follicular VEGF-A–vascular VEGFR2 axis as a main regulator for thyrotropin-dependent thyroid angiofollicular remodeling and goitrogenesis.

**Keywords** thyroid angiofollicular unit; thyroid goitrogenesis; vascular remodeling; VEGFR2
**Subject Categories** Metabolism; Vascular Biology & Angiogenesis

## Introduction

Thyroid gland is the largest endocrine organ in our body and is composed of epithelial follicles that are uniformly interfaced with fenestrated capillaries to make up "angiofollicular units" for proper secretion of thyroid hormones (Colin *et al*, 2013). Endocrine function of thyroid gland is primarily regulated by thyrotropin (thyroid-stimulating hormone, TSH) that is secreted from pituitary thyrotropes (Chiamolera & Wondisford, 2009; Brent, 2012). Circulating thyrotropin specifically activates its receptor on basolateral membrane of thyroid follicular epithelial cells (thyrocytes) (Colin *et al*, 2013). Conditions of hyper-activation of thyrotropin receptor such as Graves' disease display follicular hyperplasia and a parallel increase in capillary density in thyroid gland (Davies *et al*, 2005). Thus, alteration of thyrotropin receptor activation may modulate not only the structure and endocrine function of thyroid follicular cells but also their surrounding capillaries via regulating their secretion of angiogenic factors (Imada *et al*, 1986a,b; Davies *et al*, 2005; Colin *et al*, 2013).

1 Center for Vascular Research, Institute of Basic Science (IBS), Daejeon, Korea
2 Graduate School of Medical Science and Engineering, Korea Advanced Institute of Science and Technology (KAIST), Daejeon, Korea
3 Department of Otorhinolaryngology-Head and Neck Surgery and Biomedical Research Institute, Pusan National University School of Medicine, Pusan National University Hospital, Busan, Korea
4 Graduate School of Nanoscience and Technology, Korea Advanced Institute of Science and Technology (KAIST), Daejeon, Korea
5 Department of Pathology, Pusan National University School of Medicine, Pusan National University Hospital, Busan, Korea
6 Department of Physiology and Cell Biology, Graduate School of Medicine, Kobe University, Kobe, Japan
7 Department of Vascular Biology, The Sakaguchi Laboratory, School of Medicine, Keio University, Shinjuku-ku, Tokyo, Japan
8 Laboratory of Molecular Biology, Center for Cancer Research, National Cancer Institute, National Institutes of Health, Bethesda, MD, USA
9 Lunenfeld-Tanenbaum Research Institute, Mount Sinai Hospital, Toronto, ON, Canada
10 Department of Internal Medicine, Seoul National University College of Medicine, Seoul, Korea
11 Wihuri Research Institute and Translational Cancer Biology Program, Biomedicum Helsinki, University of Helsinki, Helsinki, Finland
12 Research Center for Endocrine and Metabolic Diseases, Chungnam National University School of Medicine, Daejeon, Korea
*Corresponding author. Tel: +82 42 280 7161; Fax: +82 42 280 7995; E-mail: minhos@cnu.ac.kr
**Corresponding author. Tel: +82 42 350 2638; Fax: +82 42 350 2610; E-mail: gykoh@kaist.ac.kr
†These authors contributed equally to this work

 

Endothelial cells (ECs) perform and maintain diverse and heterogeneous functions in different organs and tissues such as bone, retina, liver, uterus, and Schlemm's canal in eye (Aird, 2007a,b; Kim *et al*, 2013; Aspelund *et al*, 2014; Hu *et al*, 2014; Kusumbe *et al*, 2014; Park *et al*, 2014). In case of thyroid glands, each thyroid follicle is surrounded by a rich capillary network which provides substrates for hormone synthesis and transports the released hormones (Colin *et al*, 2013). The thyroid capillaries have a distinguishable characteristic of endothelial fenestrae, transcellular pores that are 60–80 nm in diameter spanned by fenestral diaphragms, which is a critical component in the vasculatures of endocrine organs for proper transport of low-molecular weight hydrophilic molecules (Esser *et al*, 1998; Risau, 1998; Kamba *et al*, 2006). The propinquity of the fenestrated endothelium to thyroid follicular epithelium suggests that these two cell types interact with each other to develop and maintain the fenestrae (Risau, 1998). However, how the fenestrae in the endothelium of thyroid are maintained and regulated needs to be finely defined.

Maintenance of thyroid vasculature integrity is known to be majorly regulated by two key vascular ligands–receptor systems: vascular endothelial growth factors (VEGFs) and their receptors (VEGFRs), and angiopoietins (Angs) and the Tie2 receptor (Carmeliet, 2003; Augustin *et al*, 2009; Lohela *et al*, 2009; Chung & Ferrara, 2011). Earlier studies indicated that VEGF-A produced by thyroid follicle cells in response to stimulators of TSHR in thyroid glands is the key factor for maintenances of vascular density and endothelial fenestrae in the thyroid glands (Sato *et al*, 1995; Viglietto *et al*, 1997; Ramsden, 2000; Kamba *et al*, 2006; Colin *et al*, 2013). Moreover, in certain conditions such as thyroid goiter, in which TSHR is hyper-activated, high expression of VEGF-A has been noted (Nagura *et al*, 2001). Furthermore, blockades of VEGF-A or Angs have been reported to alleviate experimental goitrogenesis (Ramsden *et al*, 2005). However, the mechanical insights behind how such beneficial effects occur are still lacking. Moreover, expression and roles of vascular ligands–receptors, VEGFRs and Tie2, in the thyroid capillaries need to be better defined to identify a reasonable therapeutic avenue for thyroid goiter.

In this study, we determined the expression patterns of the two vascular systems, evaluated the patterns of follicular and vascular remodeling and their key regulators under different TSH levels, and investigated the molecular pathway responsible for mediating TSHR-associated follicular remodeling and goitrogenesis.

# Results

## Profound and distinct expression of VEGF-A, VEGFR2, and VEGFR3, but no Ang1, Ang2, and Tie2, in thyroid glands

To gain insight into the role of vascular ligand–receptor system in thyroid glands, we first examined their relative expression in thyroid glands and compared those levels with other organs. Quantitative real-time PCR analyses revealed that VEGF-A mRNA expression was ~sixfold higher and PDGF-B mRNA expression was ~twofold higher, but the mRNA expression of VEGF-C, VEGF-D, angiopoietin-1 (Ang1), angiopoietin-2 (Ang2), and PDGF-A was either slightly higher, lower, or similar in the thyroid glands compared with other adjacent organs—esophagus and trachea (Fig 1A). Supporting these findings, semi-quantitative RT–PCR analyses revealed that the three major isoforms of VEGF-A—$VEGF_{120}$, $VEGF_{164}$, and $VEGF_{188}$—were highly expressed in the thyroid gland compared with other organs including lung, liver, kidney, and adrenal glands (Fig 1B). Moreover, $VEGF_{164}$ was continuously secreted by the tissue explants of the thyroid glands (Fig 1C). Consistent with these findings, reporter mice analyses revealed that VEGF-A was highly expressed in the thyroid glands, while VEGF-C was barely detectable (Fig 1D and Appendix Fig S1A). High-magnification images showed that VEGF-A expression was confined to within the follicular cells, not to the surrounding ECs (Fig 1D). In fact, compared with the adjacent organs, the thyroid gland had dense but well-arranged vasculatures that surround each thyroid follicle (Fig 1E and F). ECs of the vasculatures highly expressed VEGFR2 and moderately expressed VEGFR3 (Fig 1G), but rarely expressed Tie2 (Fig 1H), which is in contrast to a previous report (Ramsden *et al*, 2005). Of note, our analysis using reporter mice indicated that both Ang1 and Ang2 were not detectable in thyroid gland, while Ang1 was detected in megakaryocytes in bone marrow and Ang2 was detected in the tip ECs of growing retinal vessels (Fig 1H and Appendix Fig S1B and C). These data clearly indicate that Ang–Tie2 system is not the major pathway regulating thyroid angiofollicular unit. Collectively, these data suggest that the capability of transcellular transport in

**Figure 1. Robust and distinct expression of VEGF-A and VEGFR2 in the thyroid gland.**

A   Comparisons of mRNA levels of genes that regulate angiogenesis in thyroid glands and adjacent organs including trachea and esophagus. Data are presented as relative folds to the levels of trachea after normalization with β-actin. Each group, $n = 3$.
B   Semi-quantitative RT–PCR showing mRNA expression of VEGF-A. Each line indicates gene transcripts of $VEGF-A_{120}$, $VEGF-A_{164}$, and $VEGF-A_{188}$.
C   Concentration of VEGF-A in supernatants at the indicated times after explant culture of the thyroid glands. Each group, $n = 5$.
D   Images showing X-gal-stained VEGF-A in the thyroid glands (Tg) of VEGF-A$^{+/LacZ}$ mouse. In right lower panel, the tissues were immunostained with anti-CD31 antibody after X-gal staining.
E   Images showing CD31$^+$ BVs of a thyroid gland (Tg) and adjacent organs including a trachea (Tr) and an esophagus (Es). Indicated region (white dotted square) is viewed under high magnification at the right panel. DAPI staining for nuclei.
F   Image showing three-dimensional distribution of BVs in thyroid follicle.
G   Images showing VEGFR2$^+$VEGFR3$^+$ thyroid gland vasculature. Right panel, X-gal staining for VEGFR2 in the thyroid glands of VEGFR2$^{+/LacZ}$ mouse.
H   Images showing barely detectable GFP signal from the thyroid glands of Tie2-GFP, Ang1-GFP, and Ang2-GFP reporter mice. Right panel, immunofluorescence staining for Ang2.

Data information: All scale bars, 50 μm except scale bars in (E), 200 μm. Kruskal–Wallis test followed by Tukey's multiple comparison test was used in (A and C). Error bars represent mean ± s.d. in (A and C).
Source data are available online for this figure.

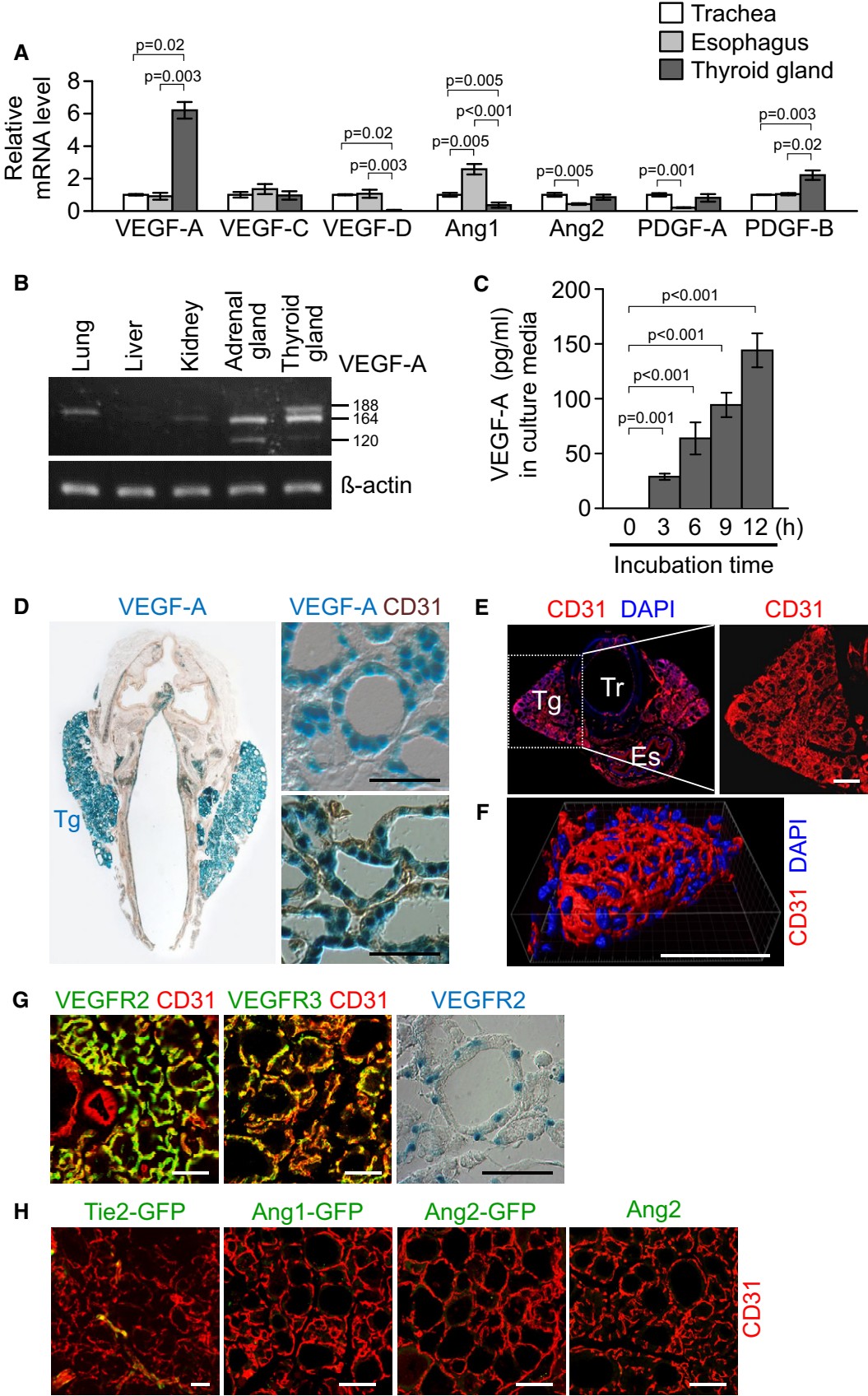

Figure 1.

thyroid glands is mainly regulated through a paracrine interaction between VEGF-A, which is continuously secreted from follicular cells, and VEGFR2- or VEGFR3-expressing ECs.

## Dynamic changes in angiofollicular unit and number of endothelial fenestrae in thyroid glands by circulating TSH

To get an insight about how the dynamic changes in thyroid capillaries occur, we determined the influence of TSHR signaling on the thyroid gland vasculature by adapting an indirect but clinically relevant mouse model (Franco *et al*, 2011) by administering levothyroxine (T4, ~0.6 mg/kg/day) or 6-propyl-2-thiouracil (PTU, ~0.72 g/kg/day) in drinking water for 3 weeks to modulate circulating TSH level. Compared with control (8.5 ± 8.1 ng/ml), T4 reduced serum TSH level (2.7 ± 1.4 ng/ml), while PTU markedly increased serum TSH level (187.6 ± 138.1 ng/ml) (Appendix Fig S2). Accordingly, shapes of follicular architectures were markedly changed by the altered serum TSH levels by administrations of T4 and PTU (Fig 2A). Compared with control, T4 reduced the height of follicular cells, vascular density, and vascular diameter in thyroid gland by 62, 31, and 20% (Fig 2A–F). In contrast, PTU increased the height of follicular cells, vascular density, and vascular diameter in thyroid gland by 94, 45, and 194% (Fig 2A–F). Of note, neither T4 nor PTU induced any apparent changes in the vasculatures of other major organs such as the submandibular gland (Fig 2E and G). PH3$^+$ proliferating ECs were often detected in the thyroid gland treated with PTU, while PH3$^+$ ECs were not or rarely detectable in the thyroid glands treated with control or T4 (Fig 2H and I), indicating that the increased EC proliferation could be involved in the PTU-induced increase in vascular density and enlargement. Moreover, intravital imaging analyses with rhodamine-conjugated RBC revealed that the PTU-induced enlarged vessels in the thyroid gland had strikingly increased blood flow (Movies EV1–EV3). Then, to determine whether the PTU-induced vascular remodeling is reversible, we examined the vasculatures 3 weeks after withdrawal of PTU administration. Intriguingly, the vascular remodeling and changes of follicular structures were almost completely restored to normal status (Appendix Fig S3A–F). Thus, not only thyroid follicles but also thyroid vasculatures are dynamically and finely adapted in response to TSH stimulation.

We next questioned whether the modulation of TSHR signaling can alter the ultrastructure of the discontinuous, fenestrated endothelium in thyroid glands. Compared with control, capillaries of thyroids treated with T4 appeared to be much thicker and had decreased number of endothelial fenestrae by 66%, whereas they showed markedly increased number of caveolae by 267% (Fig 2J and M). In contrast, PTU increased the number of endothelial fenestrae by 41% without any notable changes in fenestrae shape or caveolae number (Fig 2J and M). These ultrastructural changes were specific to thyroid glands and were undetectable in any other endocrine glands including the adrenal glands. Nevertheless, regardless of T4 or PTU, distributions and amounts of VE-cadherin, a key adherens junctional protein in EC, and NG2$^+$ pericytes were relatively well preserved for the vessel in thyroid glands (Fig 2K, L and N). These data indicate that a high level of TSH could increase the capability of transcellular transport by promoting EC content and the number of intact fenestrae on each EC, whereas a low level of TSH decreases the capability of transcellular transport by reducing EC content and the number of fenestrae on each EC in thyroid glands.

## Circulating TSH finely regulates the expression of VEGF-A, VEGFR2, and VEGFR3

Based on these findings, we further explored whether follicular VEGF-A expression is regulated by circulating TSH level using VEGF-A reporter mice (Miquerol *et al*, 2000). VEGF-A expression in the follicular cells was markedly reduced by T4 while being increased by PTU compared with control (Fig 3A). Quantitative real-time PCR analyses showed that, compared with control, the overall expression level of VEGF-A mRNA was decreased by 39% in T4-fed thyroids while being increased by 73% in PTU-fed thyroids (Fig 3B). Concomitantly, the overall concentration of VEGF-A protein was also decreased by 41% in T4-fed thyroids but increased by 181% in PTU-fed thyroids (Fig 3C), while the serum level of VEGF-A was not significantly changed under different circulating TSH levels (Fig 3D). Consistent with these findings, the mRNA level of VEGF-A in a thyroid follicular cell line, FRTL-5, was dose-dependently increased 3 hr after TSH supplementation to growth medium (Fig 3E and F). The TSH-responsive increment of VEGF-A expression (increased by 85% compared with control) was abrogated by adding LY294002, a PI3K inhibitor, but neither the protein kinase A inhibitor, KT5720, nor protein kinase C inhibitor, Go6976, had any significant effect, indicating that PI3K pathway is presumably responsible for the TSH-responsive increment of VEGF-A expression in thyroid follicular cells (Fig 3G).

We also investigated the expression of VEGFR2 and VEGFR3 under different circulating TSH levels. VEGFR2 reporter mice (Ema *et al*, 2006) analyses revealed that VEGFR2 expression in the perifollicular capillaries was reduced in T4-fed thyroids, whereas it was moderately increased in PTU-fed thyroids compared with control

**Figure 2.  Dynamic follicular and vascular remodeling in thyroid glands by circulating TSH.**

A–D    Images and comparisons of height of follicular cells and CD31$^+$ vascular density.
E       Intravital images after perfusion of FITC–dextran showing blood flow and vascular diameter of thyroid glands and submandibular glands (SMG).
F, G    Comparisons of vascular diameter in thyroid and submandibular glands.
H, I    Images and comparisons of PH3$^+$ proliferating ECs in thyroid glands. Arrows indicate CD31$^+$PH3$^+$ proliferating ECs.
J       Transmission electron micrographs showing the interface between ECs and follicular cells. Indicated region (white dotted square) is viewed under high magnification at lower panel. Arrowheads indicate endothelial fenestrae, while arrows indicate endothelial caveolae. Scale bars, 0.5 μm.
K, L    Images showing VE-cadherin distribution and NG2$^+$ pericyte coverage on CD31$^+$ BVs.
M       Comparisons of the number of endothelial fenestrae and caveolae per 10 μm vessel perimeter.
N       Comparisons of VE-cadherin expression (%) and NG2$^+$ pericyte coverage on CD31$^+$ BVs (%). Data are presented as percentage to the level of CD31 expression.

Data information: Each group, *n* = 5 in (B, D, F, G, I, and N). Each group, *n* = 3 in (M). Error bars represent mean ± s.d. Kruskal–Wallis test followed by Tukey's multiple comparison test was used in (B, D, F, G, I, M, and N). ns, not significant. Scale bars, 50 μm.

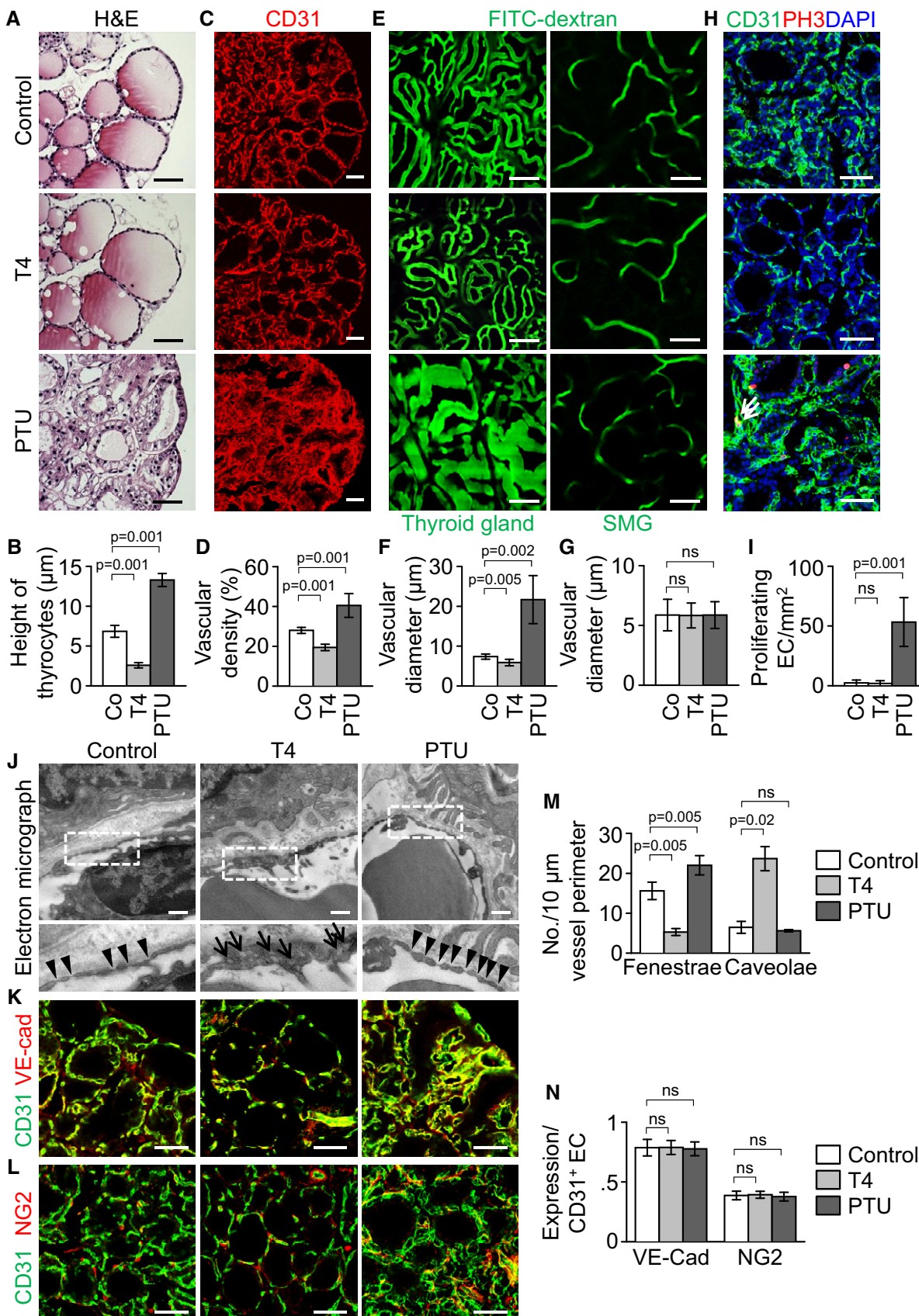

**Figure 2.**

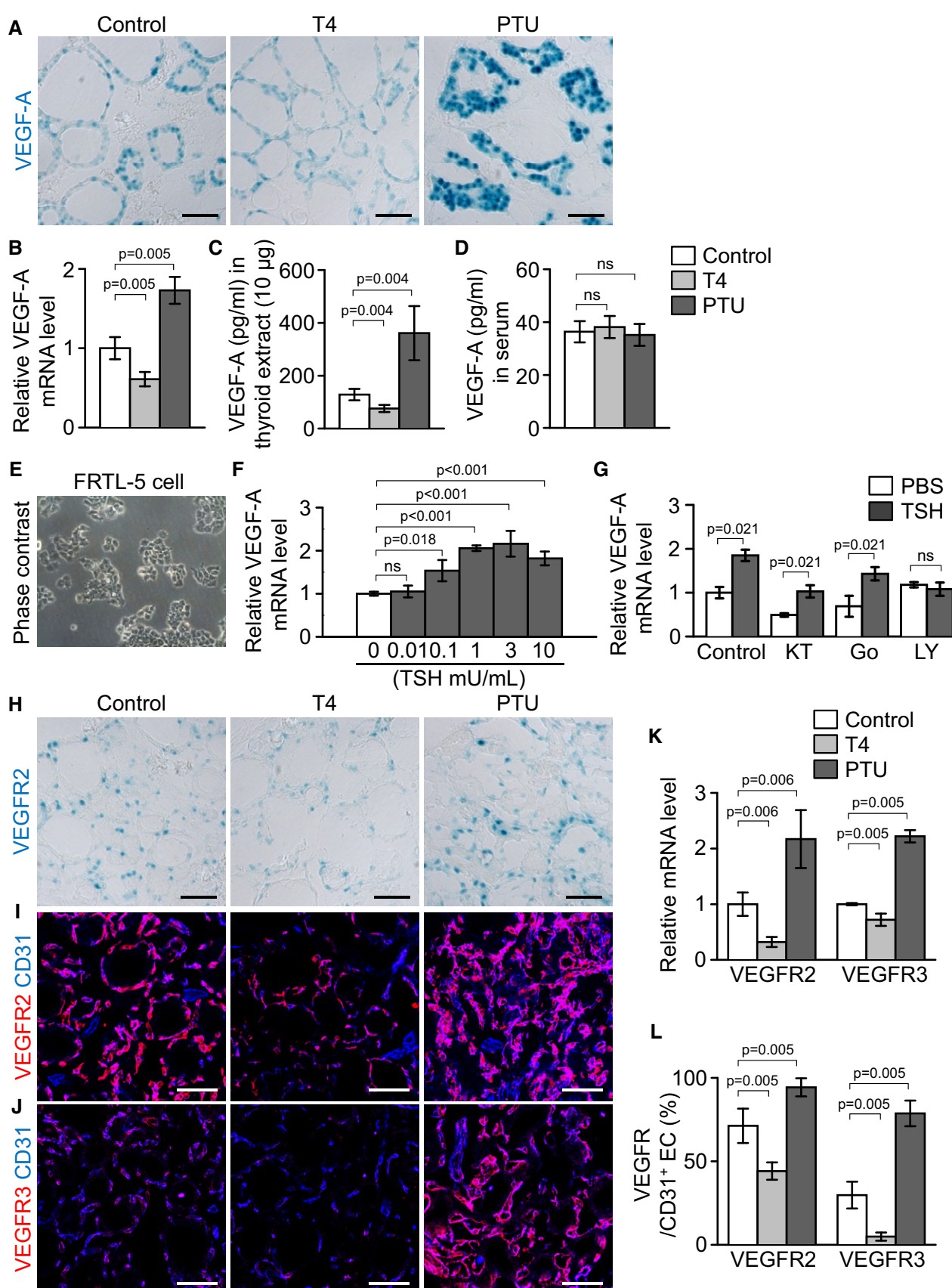

**Figure 3.**

Figure 3. Expression of follicular cell-derived VEGF-A and endothelial VEGFR is altered by different circulating TSH level.

A    Images showing X-gal-stained VEGF-A in the thyroid glands of VEGF-A[+/LacZ] mice.
B–D  Comparisons of VEGF-A mRNA and protein levels in thyroid glands, and circulating VEGF-A levels in serum.
E    Phase contrast image of cultured rat thyroid follicular cell line, FRTL-5.
F    Increases of VEGF-A mRNA expression in FRTL-5 cells at 3 h after TSH stimulation.
G    Comparisons of VEGF-A mRNA levels in FRTL-5 cells after supplement with cell signal inhibitors. Cells were pretreated for 1 h with KT5720 (KT, 3 μM), Go6976 (Go, 0.1 μM), or LY294002 (LY, 20 μM) before TSH (1 mU/ml) supplementation.
H    Images showing X-gal-stained VEGFR2 in the thyroid glands of VEGFR2[+/LacZ] mice.
I, J  Immunofluorescence images showing expression of VEGFR2 and VEGFR3.
K    Comparisons of VEGFR2 and VEGFR3 mRNA levels.
L    Comparisons of expression of VEGFR2 and VEGFR3 (relative percentage to the levels of CD31) proteins.

Data information: Each group, $n = 4$ in (B, C, D, K, and L). Each group, $n = 3$ in (F and G). Error bars represent mean ± s.d. Kruskal–Wallis test followed by Tukey's multiple comparison test was used in (B, C, D, F, K, and L). Mann–Whitney U-test was used in (G). ns, not significant. Scale bars, 50 μm.

(Fig 3H). In agreement with this finding, the overall expression level of VEGFR2 mRNA was decreased by 68% in T4-fed thyroids while being increased by 118% in PTU-fed thyroids (Fig 3K). The overall expression level of VEGFR3 mRNA showed similar results, being decreased by 28% in T4-fed thyroids while being increased by 122% in PTU-fed thyroids (Fig 3K). Fluorescence image analyses revealed that, under baseline conditions, the relative VEGFR expression was 71% for VEGFR2 and 30% for VEGFR3 on the CD31[+] EC (Fig 3I, J and L). However, T4 markedly decreased the relative expression by 38% for VEGFR2 and 84% for VEGFR3, whereas PTU increased the relative expression by 32% for VEGFR2 and by 164% for VEGFR3 (Fig 3I, J and L). Thus, thyroid follicular VEGF-A and endothelial VEGFR2 and VEGFR3 expressions are regulated proportionally in response to TSH stimulation.

In addition, we further evaluated tissue hypoxic status within the thyroid gland because VEGF-A expression has been known to be induced in response to tissue hypoxia in many other systems (Ferrara, 2004). However, tissue hypoxia, as quantified using Hypoxyprobe-1 and GLUT1 expression (an alternative hypoxia marker), was not meaningfully observed within the thyroid gland of control, T4-treated, or PTU-treated mice, while they were highly detected in the hypoxic regions of implanted glioblastoma (positive control; Appendix Fig S4).

## VEGFR2 but not VEGFR3 is indispensable for maintaining thyroid vascular integrity during adulthood

We next questioned what the roles of constitutive expression of VEGFR2 and VEGFR3 are for the maintenance of thyroid capillary integrity at normal condition. To uncover the role of VEGFR2 in thyroid capillaries, we generated VEGFR2[iΔEC] mice by mating the VEGFR2[flox/flox] mouse (Hooper et al, 2009) with the VE-Cadherin-Cre-ER[T2] mouse (Fig 4A). As control mice, we used VE-Cadherin-Cre-ER[T2] but flox/flox-negative mice among the littermates for each experiment. Repeated administrations of tamoxifen reduced expression of VEGFR2 and VEGFR3 by 83 and 72% in the ECs of thyroid vasculatures of VEGFR2[iΔEC] mice compared with those of control mice (Fig 4B–E). Vascular density and numbers of sprouting filopodia and fenestrae were reduced by 43, 97, and 87%, while numbers of apoptotic EC and caveolae were increased by 1,207 and 854% in thyroid vasculatures of VEGFR2[iΔEC] mice compared with those of control mice (Fig 4F–M). Electron micrograph (EM) images showed that the cytoplasmic membrane of ECs of VEGFR2[iΔEC] mice were thickened and nonfenestrated (Fig 4L), implying that the capability of transcellular transport by the ECs of thyroid

vasculatures was markedly interrupted by the VEGFR2 depletion. However, no apparent differences in gross features or vascular density, network, and branching between the retinal vessels of control and VEGFR2[iΔEC] mice were detected (Appendix Fig S5), indicating that role of VEGFR2 in the vessels is organ-specific.

To unveil the role of VEGFR3 in thyroid capillaries, we generated VEGFR3[iΔEC] mice by crossing the VEGFR3[flox/flox] mouse (Haiko et al, 2008) with the VE-Cadherin-Cre-ER[T2] mouse (Fig 5A). Repeated administrations of tamoxifen reduced the expression of VEGFR3 by 98%, but no changes in VEGFR2 expression were observed in the ECs of thyroid vasculatures of VEGFR3[iΔEC] mice compared with those of control mice (Fig 5B–F). However, no apparent differences in gross features, vascular density, and numbers of apoptotic EC and EC fenestrae were detected between the thyroid glands of control and VEGFR3[iΔEC] (Fig 5G–L). Thus, the role of VEGFR2 is indispensable, but that of VEGFR3 is dispensable to maintain the integrity of ECs of thyroid capillaries including endothelial fenestrae.

## VEGF-A is required for maintenance of vasculature integrity and PTU-induced angiofollicular remodeling in thyroid glands

We further evaluated the role of VEGF-A in adult normal thyroid glands by using a VEGF-Trap (Holash et al, 2002), which has been known to block VEGF-A, VEGF-B, and placental growth factor. Compared with control, administration of VEGF-Trap (intraperitoneal injections of 25 mg/kg every 2 days for 2 weeks) reduced the vascular density by 36%, regressed capillary beds, increased the number of apoptotic ECs, but preserved NG2[+] pericyte coverage (Fig 6A–F), indicating that VEGF-A plays a substantial role in capillary maintenance in adult thyroid glands. Furthermore, VEGF-Trap reduced the number of fenestrae by 66%, whereas it increased the number of caveolae by 242% (Fig 6G and H), indicating that the capability of transcellular transport in thyroid ECs can also be seriously impaired by VEGF-A blockade. Interestingly, fluorescence image analyses indicated that the expression of VEGFR2 and VEGFR3 was also decreased by 36 and 83%, respectively, in thyroid capillaries after VEGF-Trap treatment (Fig 6I–L), noting that constitutive stimulation of VEGFR2 in thyroid ECs by follicular VEGF-A is required for the maintenance of thyroid vasculatures.

We next explored whether VEGF-A blockade affects PTU-induced angiofollicular remodeling which mimics thyroid goiter. Starting from day 7 after PTU administration by drinking water, the mice received intraperitoneal injections of VEGF-Trap (25 mg/kg) every 2 days for the subsequent 2 weeks. Compared with control, administrations of PTU increased the vascular density by 53% and

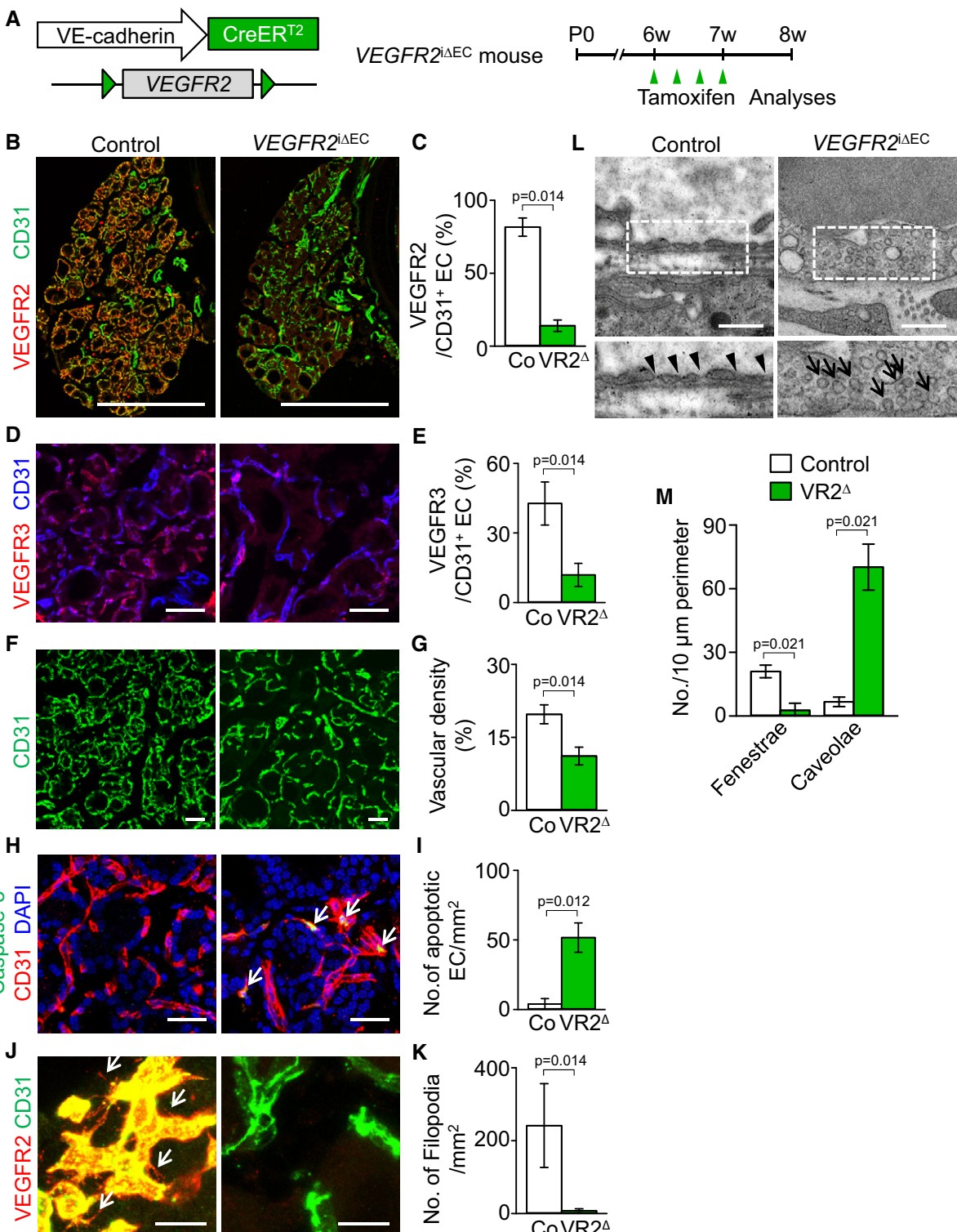

**Figure 4.  Inducible genetic deletion of VEGFR2 leads to vessel regression and loss of endothelial fenestrae in the thyroid gland.**

A    Diagram for EC-specific depletion of VEGFR2 in the thyroid vessels of 6-week-old *VEGFR2*iΔEC mice and their analyses 2 weeks later.

B–M  Images and comparisons of (B–E) expression of VEGFR2 and VEGFR3 in CD31+ ECs, (F, G) CD31+ vascular density, (H, I) numbers of caspase-3+ apoptotic EC (white arrows), (J, K) filopodia/sprouting (white arrows), and (L, M) fenestrae (black arrowheads) and caveolae (black arrows) in the thyroid capillaries of control littermates (Co; n = 4–5) and *VEGFR2*iΔEC mice (VR2Δ; n = 4–5).

Data information: All scale bars, 50 μm except scale bars in (B), 500 μm, and scale bars in (L), 0.5 μm. Error bars represent mean ± s.d. Mann–Whitney *U*-test was used in (C, E, G, I, K, and M).

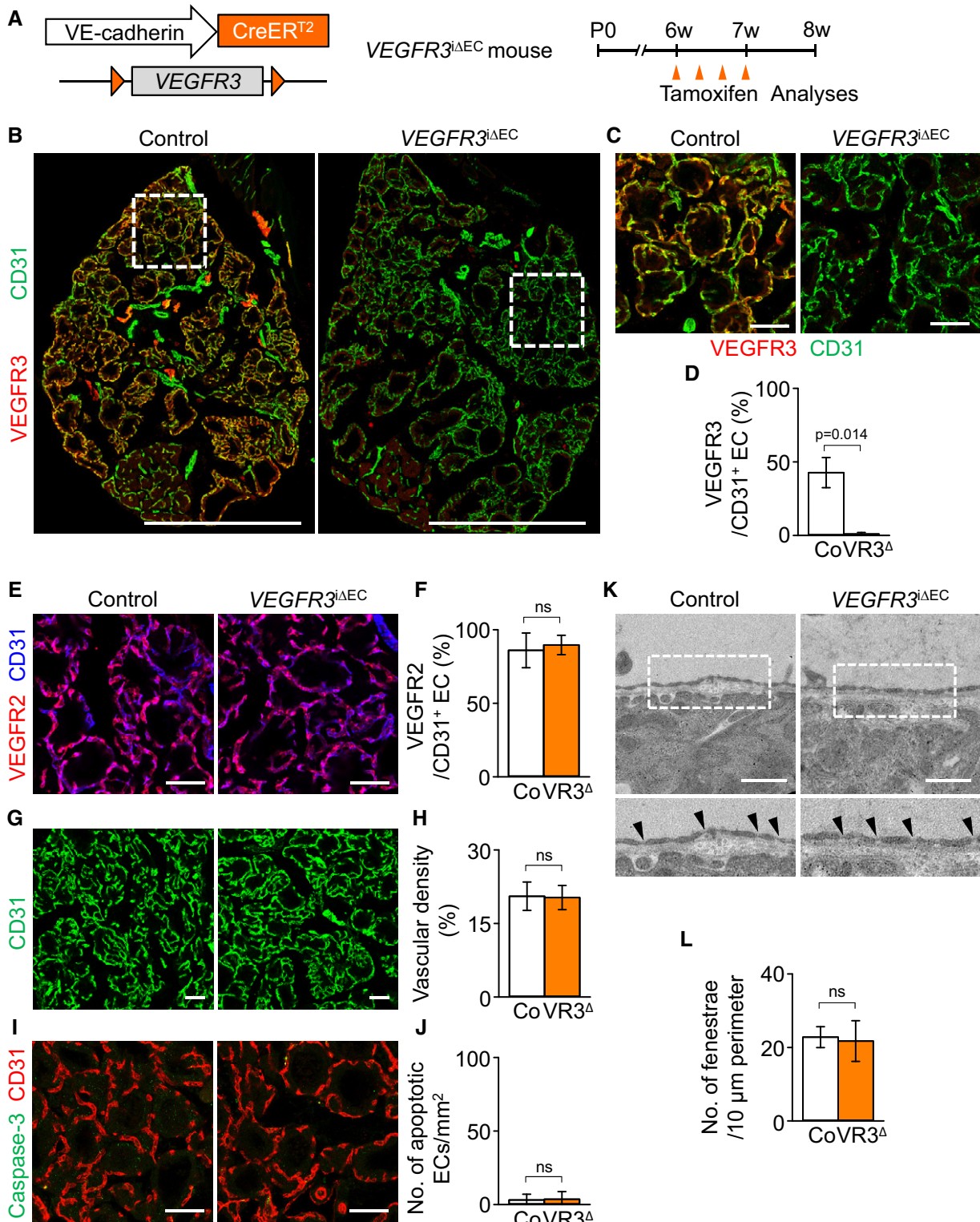

**Figure 5.  Inducible genetic deletion of VEGFR3 does not induce phenotypic changes of thyroid gland vasculatures.**

A    Diagram for EC-specific depletion of VEGFR3 in the thyroid vessels of 6-week-old *VEGFR3*[iΔEC] mice and their analyses 2 weeks later using *VEGFR3*[iΔEC] mice.

B    Images of whole thyroid glands in *VEGFR3*[iΔEC] mice and their littermate controls. Indicated region (box) is viewed under high magnification as (C). Scale bars, 500 μm.

C–L  Images and comparisons of (C–F) expression of VEGFR3 and VEGFR2 in CD31[+] ECs, (G, H) CD31 vascular density, (I, J) numbers of caspase-3[+] apoptotic ECs, and (K, L) fenestrae (black arrowheads) in the thyroid capillaries of control littermates (Co; *n* = 4–5) and *VEGFR3*[iΔEC] mice (VR3[Δ]; *n* = 4–5).

Data information: All scale bars, 50 μm except scale bars in (K), 0.5 μm. Error bars represent mean ± s.d. Mann–Whitney *U*-test was used in (D, F, H, J, and L).

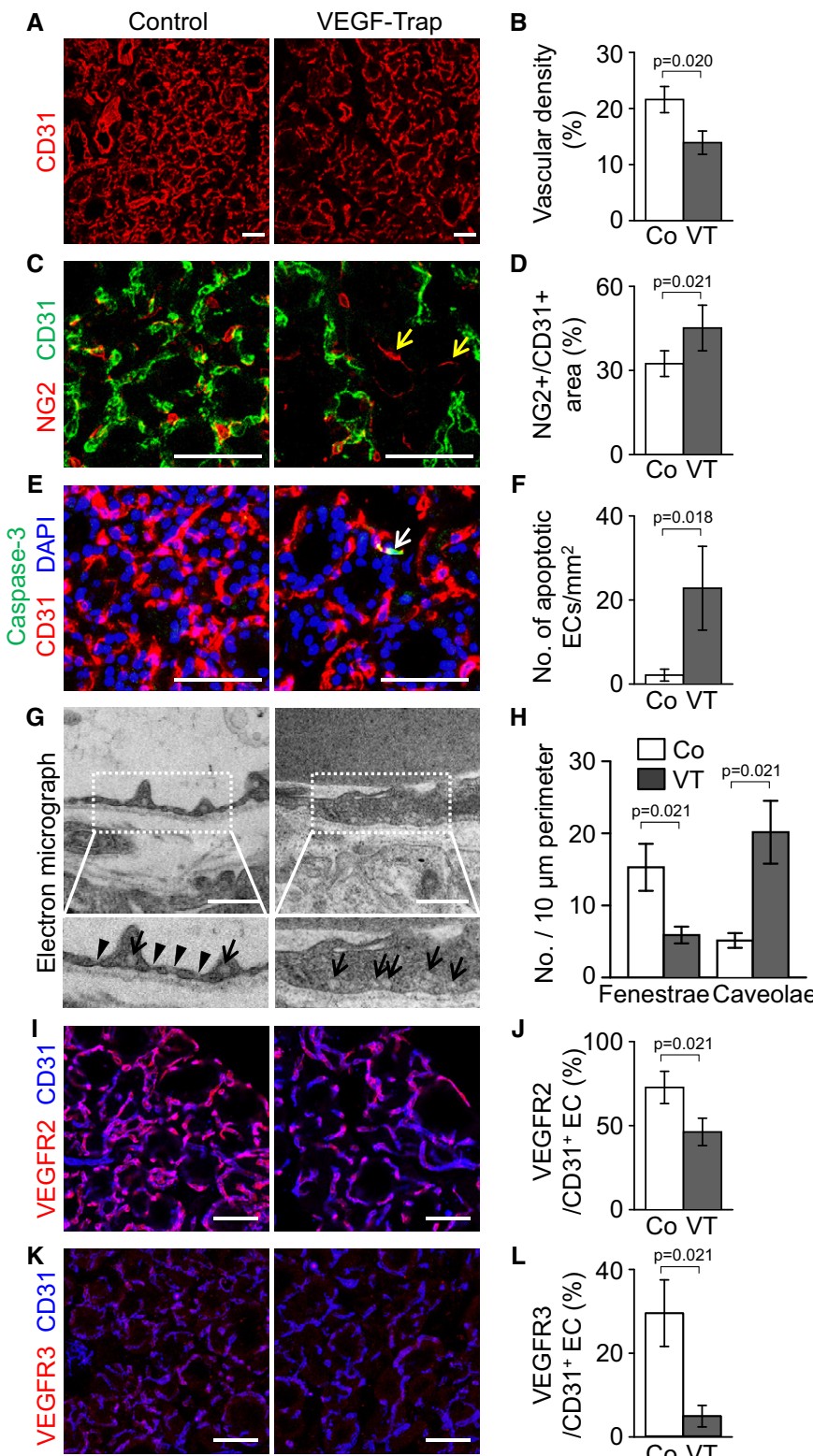

**Figure 6.  VEGF-A blockade causes vessel regression and reduction of endothelial fenestrae in thyroid gland.**

A–L    Images and comparisons of (A, B) CD31[+] vascular density, (C, D) NG2[+] pericyte coverage on CD31[+] BVs, nascent NG2[+] pericytes on the regressed ECs (yellow arrows), (E, F) numbers of caspase-3[+] apoptotic EC (white arrow), (G, H) fenestrae (arrowheads) and caveolae (arrows) in endothelium, and (I–L) expression of VEGFR2 and VEGFR3 in CD31[+] ECs of control (Co; n = 4–5) or VEGF-Trap-treated mice (VT; n = 4–5).

Data information: All scale bars, 50 μm except scale bars in (G), 0.5 μm. Error bars represent mean ± s.d. Mann–Whitney U-test was used in (B, D, F, H, J, and L).

vascular diameter by 161%, once again validating that PTU-induced vascular remodeling leads to vascular enlargement in thyroid glands (Appendix Fig S6A–D). However, VEGF-Trap treatment markedly abrogated this PTU-induced effect, reducing the vascular density and vascular diameter by 38 and 64%, respectively, almost completely restoring the thyroid vasculature to its normal state (Appendix Fig S6A–D). Regardless of the predominant outcomes of PTU-induced vascular remodeling, the vasculature of VEGF-Trap-injected mice had well-organized morphology, even in high levels of TSH, indicating that follicular VEGF-A is responsible for the dynamic vascular remodeling by high TSH level. Strikingly, administration of VEGF-Trap markedly decreased the increased heights of PTU-fed thyroid follicular cells by 64%, becoming very similar to those of normal mice (Appendix Fig S6E and F). Furthermore, increments of the thyroid weights after PTU administration were strikingly decreased by VEGF-Trap injection (Appendix Fig S6G and H).

### Genetic depletion of VEGFR2 suppresses PTU-induced thyroid angiofollicular remodeling and reduces goiter weights

To determine the role of VEGFR2 in the PTU-induced thyroid angio-follicular remodeling, we administered PTU by drinking water for 3 weeks to ~6-week-old $VEGFR2^{\Delta iEC}$ mice (Fig 7A). As control mice, we used $VE\text{-}Cadherin\text{-}Cre\text{-}ER^{T2}$ but flox/flox-negative mice among the littermates for each experiment while PTU was administered for 3 weeks. Markedly reduced expression of VEGFR2 (84%) and VEGFR3 (68%) was detected in the ECs of thyroid vasculatures of $VEGFR2^{\Delta iEC}$ mice by repeated administrations of tamoxifen compared with those of control mice (Fig 7B–E). Strikingly, the PTU-induced increases of thyroid vascular density, number of EC fenestrae, heights of follicular cells, and thyroid weights were reduced by 75, 82, 57, and 79%, respectively, in $VEGFR2^{\Delta iEC}$ mice compared with those of control mice (Fig 7F–M). Of note, EM images revealed strikingly increased number of endothelial caveolae by 1,444% and thickened cytoplasmic membrane of ECs in the $VEGFR2^{\Delta iEC}$ mice (Fig 7H and I). These findings demonstrate that the PTU-induced vascular remodeling and goitrogenesis is mostly dependent on VEGFR2 in the ECs of thyroid vasculatures. Long-term depletion of VEGFR2 impairs the integrity of thyroid vasculatures.

### VEGFR2 neutralizing antibody DC101 attenuates the PTU-induced changes of angiofollicular remodeling, goiter weight, and expression of TSH-responsive genes in thyroid glands

We further performed the experiment using DC101, a VEGFR2 neutralizing antibody, to see whether antibody-mediated VEGFR2 blockade can reduce the PTU-induced angiofollicular remodeling as well as goiter weights. Starting from day 7 after PTU administration by drinking water, the mice received intraperitoneal injections of DC101 (25 mg/kg) every 2 days for the subsequent 2 weeks. In comparison with Fc treatment, the vessel densities in the thyroids were decreased by 43% after DC101 administration compared with Fc-treated thyroids during PTU-induced goitrogenesis (Fig 8A and E). Notably, we found that, compared with those of Fc treatment, the heights of follicular cells were 65% less (Fig 8B and F), and the thyroid weight was also reduced by 68% after DC101 administration in PTU-fed mice (Fig 8C and G), both being restored very closely to the levels of normal mice. Nevertheless, DC101 administration did

not significantly alter the elevated circulating TSH levels during the PTU-induced goitrogenesis (Fig 8H). Compared with Fc administration, administration of DC101 for 2 weeks strikingly reduced the endothelial fenestrae by 80% and increased endothelial caveolae by 298% in PTU-fed mice (Fig 8D, I and J). Consequently, we next questioned whether the blockade of VEGFR2 would decrease the expression levels of TSH-responsive genes in thyroid glands. Based on previous reports (Gerard et al, 1988; Garcia & Santisteban, 2002; Mascia et al, 2002), we selected three TSH-responsive genes, transcription factor Pax8, thyroid peroxidase (TPO), and sodium/iodide symporter (NIS), and performed quantitative real-time PCR analyses in the thyroid glands after PTU with Fc or DC101 administration. As expected, administration of PTU plus Fc significantly increased the mRNA expression levels of Pax8 (~128-fold), TPO (~251-fold), and NIS (~4,643-fold) in the thyroid glands compared with control mice (Fig 8K). However, administration of PTU plus DC101 reduced the mRNA level of TSH-responsive genes by 57% in Pax8, 43% in TPO, and 55% in NIS compared with those of the PTU-fed thyroids (Fig 8K). Thus, the reduction in thyroid EC fenestration by the blockade of VEGFR2 ultimately diminished the TSH responsiveness of follicular cells during PTU-induced goitrogenesis. Together, an antibody-mediated blockade of VEGFR2 can abrogate vascular remodeling as well as hypertrophy of follicular cells, leading to the reduction of thyroid weight during PTU-induced goitrogenesis.

### Microvessel density and VEGFR2 expression are increased in thyroid glands of patients with Graves' disease

The experimental model of PTU-induced thyroid goiter shares similar features with human Graves' disease, including thyrotropin receptor hyper-stimulation and follicular hypertrophy. Thus, to validate increases of VEGFR2 expression in human pathology, thyroid specimens of Graves' disease were examined in comparison with normal thyroid glands as a control. Immunostaining of CD31 and VEGFR2 showed confined expression of VEGFR2 within CD31$^+$ BVs, not within follicular cells of normal or Graves' disease-thyroids (Fig 9A and B). CD31$^+$ BV density was significantly increased in Graves' disease-thyroid glands compared with control (Fig 9A and C). Importantly, the VEGFR2 immunostaining scores, measured as a sum of signal intensity of immunoreactions and proportion of VEGFR2 expression on CD31$^+$ BVs, were strikingly increased compared with control (Fig 9B and D). Thus, microvessel densities and their VEGFR2 expression were increased in the patient with Graves' disease.

## Discussion

In this study, we found profound and distinct expression of VEGF-A, VEGFR2, and VEGFR3, but no Ang1, Ang2, and Tie2, in thyroid glands. Importantly, using genetically modified mice and selective blockades of VEGF-A or VEGFR2, we uncovered the indispensable roles of VEGF-A and VEGFR2 and dispensable role of VEGFR3 in the ECs of thyroid capillaries for maintenance of integrity including endothelial fenestrae. In addition, this study demonstrates that blockade of VEGFR2 markedly ameliorates hypertrophy of follicular cells as well as vascular remodeling, leading to amelioration of PTU-induced goitrogenesis.

Because inhibiting VEGF-A–VEGFR2 pathway affects not only its main target of thyroid vasculatures but also neighboring follicular

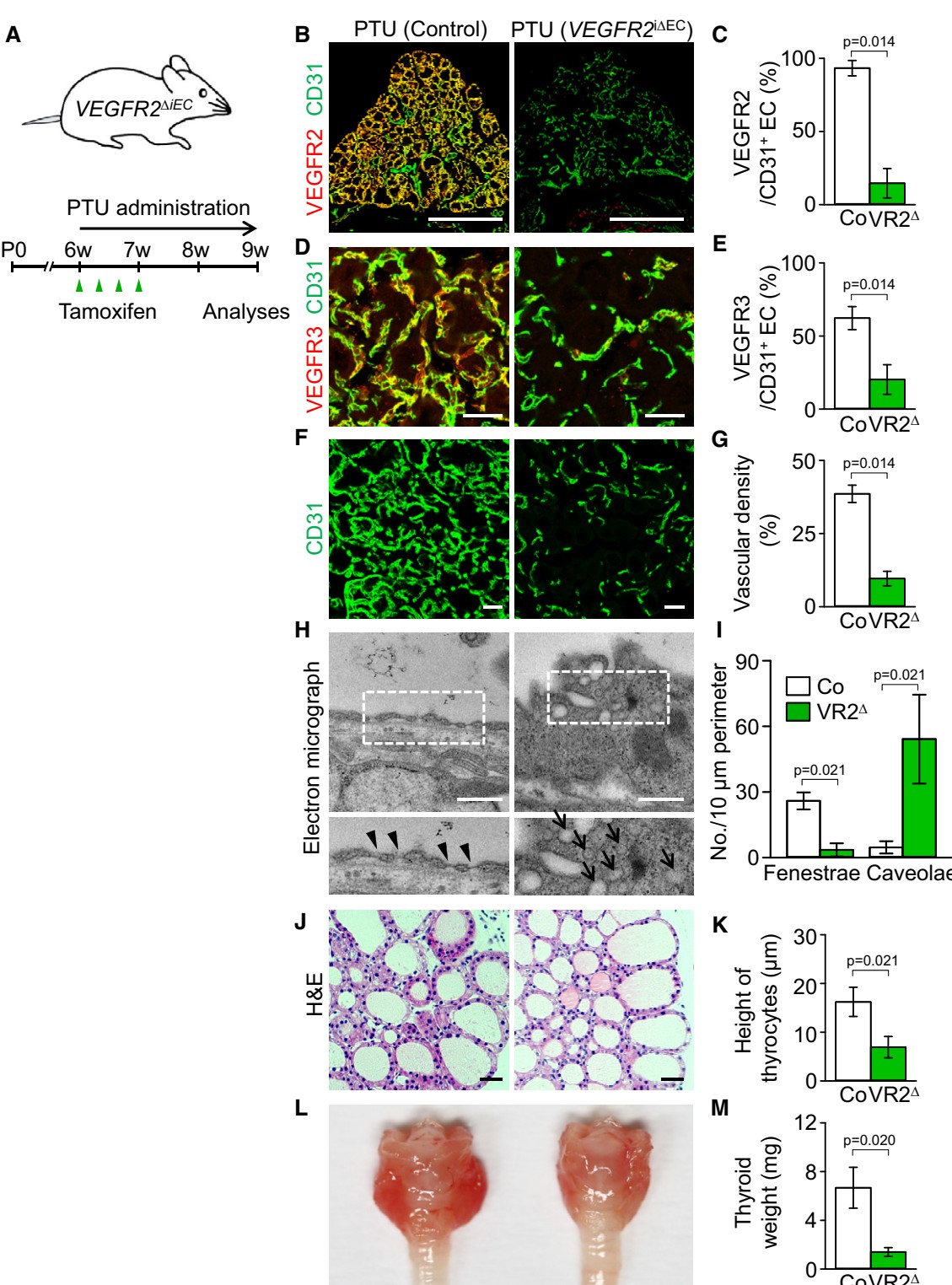

**Figure 7. Inducible genetic deletion of VEGFR2 ameliorates thyroid angiofollicular remodeling during goitrogenesis.**

A   Diagram showing PTU administration into *VEGFR2*[iΔEC] mice.

B–K   Images and comparisons of (B–E) expression of VEGFR2 and VEGFR3, (F, G) CD31[+] BV density, (H, I) fenestrae (arrowheads) and caveolae (arrows), and (J, K) height of thyrocytes in *VEGFR2*[iΔEC] (VR2[Δ]) and their littermate controls (Co) after 3 weeks of PTU administration. All scale bars, 50 μm except scale bars in (B), 500 μm, and scale bars in (H), 0.5 μm.

L, M   Gross images and comparisons of thyroid weights after 3 weeks of PTU administration in *VEGFR2*[iΔEC] mice compared with Co. Scale bar, 1 mm.

Data information: Each group, *n* = 4–5. Error bars represent mean ± s.d. Mann–Whitney *U*-test was used in (C, E, G, I, K, and M).

cells, we speculated that decreased transcellular permeability through the inhibition of VEGF-A–VEGFR2 pathway may inhibit TSH from reaching follicular cells. Consistently, blockade of VEGFR2 was sufficient to result in loss of endothelial fenestrae with decreases of TSH-responsive genes in thyroid glands, despite the high circulating TSH levels during PTU-induced goitrogenesis. Thus, thyroid VEGF-A–VEGFR2 pathway has a dual role, being responsible for TSH-dependent vascular remodeling and maintenance of TSH responsiveness of follicular cells by modulating endothelial fenestrae. Our findings highlight the importance of thyroid vasculature that coordinates with its neighboring follicular cells to accomplish proper hormonal feedback regulation.

We also delineated the phenotypic characterization and molecular pathways that contribute to follicular and vascular remodeling in thyroid glands under different circulating TSH level conditions in mice. High level of TSH strongly stimulated thyroid follicular cells through TSHR, and in turn the follicular cells massively produced and secreted VEGF-A to activate VEGFR2 on neighboring ECs in a paracrine manner. As a consequence, there was not only profound vascular remodeling that was mainly characterized by vascular enlargement, increased blood flow, and increases of endothelial fenestrae, but also hypertrophy of follicular cells. Thus, thyroid vascular flow and permeability are finely regulated to meet the endocrine requirements as regulated by TSH stimulation.

Thyrotropin receptor hyper-activation has been known to be associated with human thyroid diseases, such as Graves' disease, thyroid goiter, and tumor (Davies et al, 2005). Given that the stimulation of thyrotropin receptor is the main mechanism of Graves' disease, our PTU-induced mouse model of thyroid goiter shares similar features with Graves' disease, such as thyrotropin receptor hyper-stimulation, hypertrophy of thyrocytes, and distinguishable characteristics of vascular remodeling (Nagura et al, 2001; Davies et al, 2005). Intriguingly, we found increased microvessel density and their VEGFR2 expression in the thyroid glands of patient with Graves' disease. Moreover, administration of VEGFR2 blocking antibody, DC101, ameliorated PTU-induced follicular and vascular remodeling and reduced thyroid weights nearly to the level of normal mice. We suggest that these were achieved through decreased transcellular transport capacities due to decreased vascularity and endothelial fenestrations. It remains to be studied whether decreased transcellular transport capacities through blockade of VEGF-A or VEGFR2 will be able to reduce the contacts of circulating IgG antibodies to follicular cells in Graves' disease. Nevertheless, inhibitors of VEGF-A–VEGFR2 signaling could reduce the sizes of thyroid goiter, thus providing a novel therapeutic concept of

targeting vasculatures in these TSHR-associated diseases such as Graves' disease and warranting further clinical investigation.

Our findings indicate that, like other endocrine glands, thyroid glands highly express VEGF-A, which is a critical molecule to sustain the central thyroid function, release of thyroid hormones. This fact may be an important clue for understanding the recent problems in clinic regarding hypothyroidismal complications after anti-VEGF therapy (Chan et al, 2007; van der Veldt et al, 2013; Cao, 2014). Hypothyroidism was first reported in patients receiving sunitinib, a small-molecule multiple tyrosine kinase inhibitor that also affects VEGFRs (Desai et al, 2006; Rini et al, 2007). Moreover, another anti-angiogenic agent, bevacizumab, has been reported to elicit reduced thyroid perfusion with elevated TSH level (van der Veldt et al, 2013). Recently, a preclinical research comprehensively evaluated the degree of vascular alterations in healthy mouse tissues in response to VEGF-A and VEGFR2 blockades and reported that, among all analyzed tissues and organs, the vasculatures of endocrine organs were most heavily affected, particularly the thyroid (Kamba et al, 2006; Yang et al, 2013; Zhang et al, 2016). Our analyses using reporter mouse system might provide possible explanations for the anti-VEGF therapy-induced hypothyroidism. Supporting these reports, our results indicate that the decreases of endothelial fenestrae as well as vascular densities are one of the responsible causes of thyroid dysfunction after anti-VEGF therapy. Nevertheless, several blocking antibodies and small molecules for VEGF-A–VEGFR2 signaling (Hurwitz et al, 2004; Van Cutsem et al, 2012; Wilke et al, 2014) are currently being used for patients with solid cancers such as colorectal cancers. Therefore, it would be ideal to follow up on the progression of thyroid goiter for those who concurrently have cancer and thyroid goiter and are being treated with such blocking agents.

One intriguing aspect of this study is the significant expression and change of VEGFR3 in the ECs of quiescent and growing thyroid vasculatures, similar to what was observed in growing retinal vasculatures (Benedito et al, 2012). Considering the observed expression patterns of VEGFR3 ligands (VEGF-A, high; VEGF-C, low) in our study, and since it is already known that VEGFR2 and VEGFR3 react with VEGF-A by forming a homodimer and heterodimer with VEGFR2, respectively (Planas-Paz et al, 2012), we speculated that VEGFR3 might critically contribute to regulating the number and shape of fenestrae in the endothelium of thyroid vasculature. However, the analysis on the VEGFR3$^{\Delta iEC}$ mice did not meaningfully meet our speculation. Thus, VEGFR3 may play a dispensable role, together with VEGFR2, in maintaining fenestration of endothelium in the thyroid gland. These results appear to be discrepant with a previous observation that genetic depletion of endothelial VEGFR3 induces hypersprouting of growing retinal

**Figure 8. Antibody-mediated VEGFR2 blockade ameliorates thyroid hypertrophy by reducing thyrotropin responsiveness during goitrogenesis.**

A, B    Images showing CD31$^+$ BVs and H&E-stained thyrocytes in mice treated with control (Co), PTU (for 3 weeks), PTU plus Fc (for 2 weeks), or PTU plus DC101 (DC, for 2 weeks). Scale bars, 50 μm.

C    Gross images of thyroid glands. Scale bar, 1 mm.

D    Transmission electron micrographs showing the interface between ECs and thyrocytes. Each indicated region (box) is magnified and shown as lower panel. Arrowheads, endothelial fenestrae; arrows, endothelial caveolae. Scale bars, 0.5 μm.

E–J    Comparisons of (E) CD31$^+$ BV density, (F) height of thyrocytes, (G) thyroid weight, (H) serum TSH level, and (I, J) number of endothelial fenestrae and caveolae per 10 μm vessel perimeter in control mice (Co), or mice treated with PTU, PTU plus Fc (Fc), or PTU plus DC101 (DC). Each group, $n$ = 4–7. Error bars represent mean ± s.d. Kruskal–Wallis test followed by Tukey's multiple comparison test was used.

K    Comparisons of relative mRNA levels of TSH-responsive genes including Pax8, TPO, and NIS in thyroid glands. Each group, $n$ = 3–4. Error bars represent mean ± s.d. $P$ < 0.05 versus Co. Kruskal–Wallis test followed by Tukey's multiple comparison test was used.

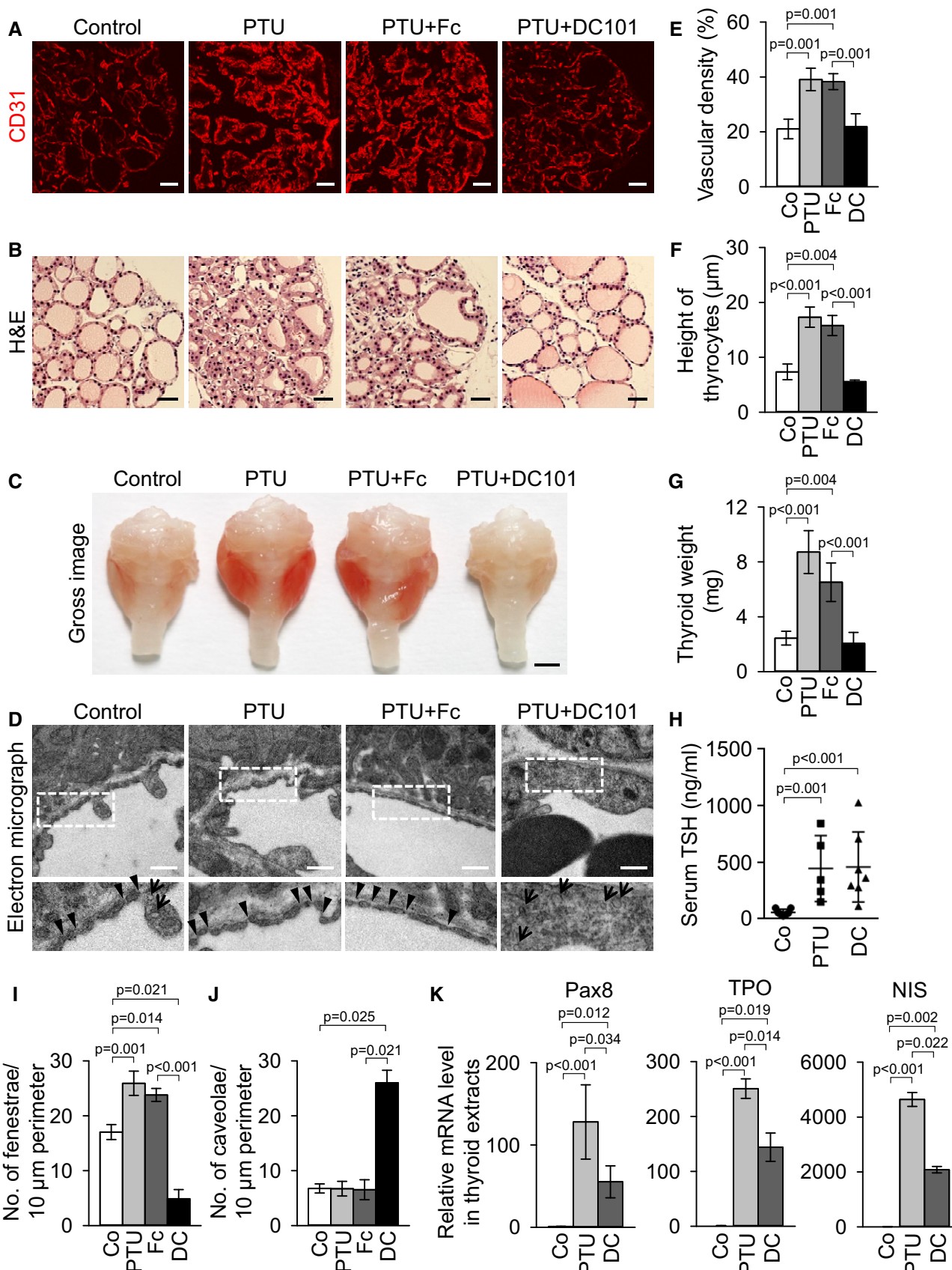

**Figure 8.**

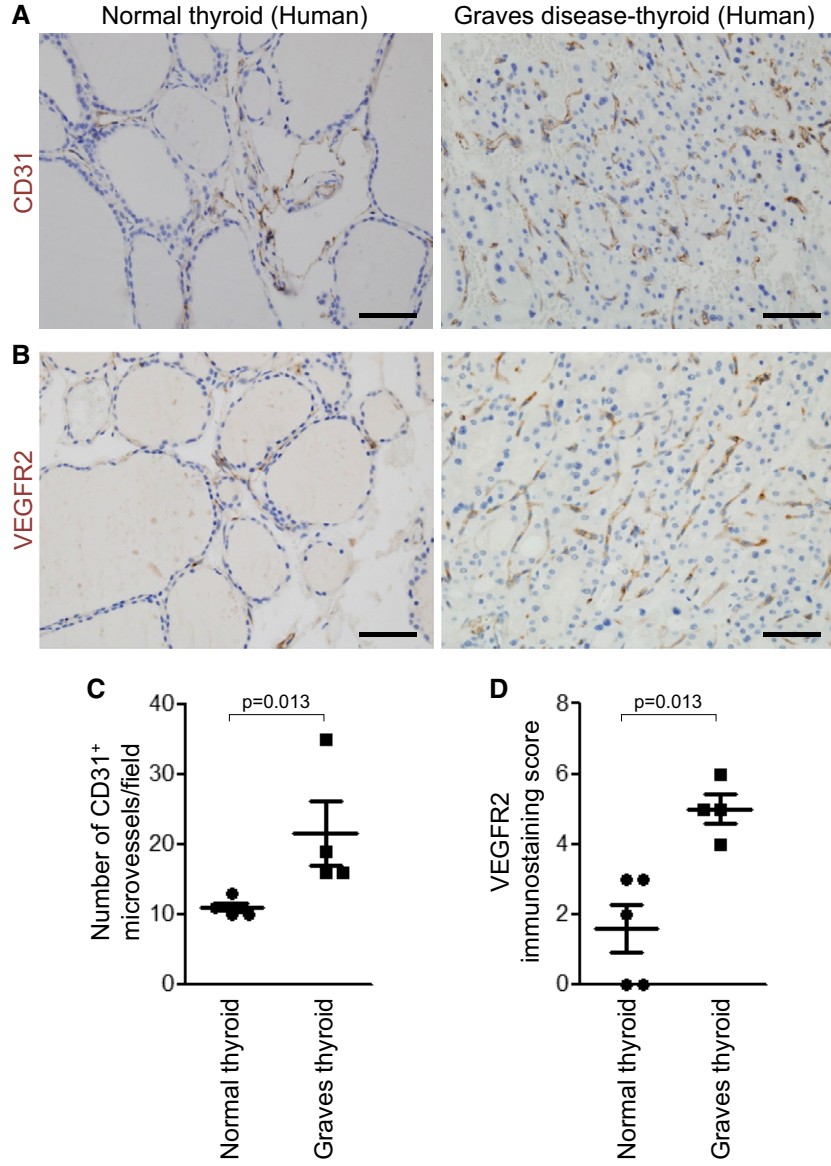

**Figure 9.  Microvessel density and VEGFR2 expression in human thyroid specimens of Graves' disease.**

A, B    Images showing CD31[+] BVs and VEGFR2 expression in normal and Graves' disease-thyroids. Scale bars, 50 μm.

C, D    Comparisons of number of CD31[+] microvessels per high-magnification field (×400) and VEGFR2 immunostaining score in normal Graves' disease-thyroids. Three areas with the highest concentration of BVs were selected and evaluated in each sample. Error bars represent mean ± s.d. Mann–Whitney *U*-test was used.

vessels (Tammela *et al*, 2011). While the previous study evaluated growing retinal vessels during postnatal development, our model system involved quiescent thyroid vasculatures during adulthood. Thus, VEGFR3 may play an indispensable role in thyroid vasculature during development period, which requires further evaluation. Moreover, since TSHR can be expressed in vascular EC (Balzan *et al*, 2012; Tian *et al*, 2014), TSH–TSHR signaling may also directly regulate the expression or signaling of VEGFR2 and VEGFR3 in thyroid vasculatures, which also needs to be further investigated.

In conclusion, we show that the vasculature of thyroid glands is finely adapted in response to TSHR stimulation and that paracrine VEGF-A–VEGFR2 pathway plays a pivotal role in maintaining transcellular transport capacities through endothelial fenestrations, thereby maintaining TSH responsiveness of follicular cells during normal state and goitrogenesis. Our findings provide an integrated view of vascular regulation in endocrine system and suggest VEGFR2 blockade as a novel therapeutic avenue for targeting vasculatures in TSHR-associated thyroid goiter.

# Materials and Methods

### RT–PCR and quantitative real-time PCR analyses

Total RNA was extracted from the indicated tissues using RNeasy-Plus Mini Kit (Qiagen) according to the manufacturer's instructions.

The RNA (2 μg) was reverse-transcribed into cDNA using Super-Script II Reverse Transcriptase (Invitrogen). Semi-quantitative RT–PCR was performed using NobleZyme™ Taq (Noble Bio) with the indicated primers. Quantitative real-time PCR (qPCR) was performed with the indicated primers using Bio-Rad™ CFX96 Real-Time PCR Detection System (Bio-Rad). The real-time PCR data were analyzed with Bio-Rad CFX Manager Software (Bio-Rad). Primers for the semi-quantitative RT–PCR (Semi) and quantitative real-time PCR analysis are indicated in Appendix Table S1.

### Quantitative measurement for VEGF-A protein

On the indicated days, thyroid glands were harvested and homogenized in ice-cold buffer containing a protease inhibitor cocktail. Serums were obtained using a standard method. To evaluate the secretory capacity of VEGF-A, an explant culture of mouse thyroid gland was performed. Briefly, one lobe of thyroid gland in each mouse was prepared by a gentle dissection and cultured in a 96-well plate (SPL Life Sciences) containing 100 μl of 6H medium consisting of Coon's modified F-12 supplemented with 5% FBS. Concentrations of VEGF-A protein in 10 μg of thyroid gland extract, 10 μl of serum, and 50 μl of culture medium were measured using the mouse VEGF-A ELISA kit (R&D systems) according to the manufacturer's protocol.

### Mice

Animal care and experimental procedures were performed under the approval from the Animal Care Committee of KAIST (KA2011-21). All procedures and animal handlings were performed following the ethical guidelines for animal studies. C57BL/6J and Tie2-GFP (FVB/N) mice were purchased from the Jackson Laboratory (Bar Harbor, ME). Ang2-EGFP mouse was purchased from the Mutant Mouse Regional Resource Centers. Ang1-GFP mouse (Zhou *et al*, 2015) was provided by Prof. Sean J. Morrison (University of Texas Southwestern Medical Center, Dallas, USA). VEGF-A$^{+/LacZ}$ mouse (Miquerol *et al*, 2000), VEGFR2$^{+/LacZ}$ mouse (Ema *et al*, 2006), VEGF-C$^{+/LacZ}$ mouse (Karkkainen *et al*, 2004), *VE-Cadherin*-Cre-ER$^{T2}$ mouse (Okabe *et al*, 2014), *VEGFR2*$^{flox/flox}$ mouse (Hooper *et al*, 2009), and *VEGFR3*$^{flox/flox}$ mouse (Haiko *et al*, 2008) were transferred and bred in specific pathogen-free animal facilities at KAIST. VEGF-A$^{+/LacZ}$ mice (Miquerol *et al*, 2000) that have elevated VEGF expression by 3′UTR insertion of the IRES-*lacZ*-pA sequence, resulting in increased stability of the VEGF-A transcripts, were used in this study to examine the expression patterns of VEGF-A since they showed almost similar vascular structures and densities in the thyroid glands compared with those of wild-type mice. The endothelial-specific *VEGFR2* or *VEGFR3* knockout mouse line was generated by crossing the *VEGFR2*$^{flox/flox}$ mouse or *VEGFR3*$^{flox/flox}$ mouse with *VE-Cadherin*-Cre-ER$^{T2}$ mouse. For all the experiments involving *VEGFR2* or *VEGFR3* knockout mouse, the control mice were *VE-Cadherin*-Cre-ER$^{T2}$ but flox/flox-negative mice among the littermates for each experiment. All mice used were males (6- to 8-week-old) and maintained on a C57BL/6J strain background. The number of mice used is indicated in each figure. The mice were assigned to experimental groups based on their genotype. Tamoxifen (Sigma-Aldrich) was dissolved in corn oil (Sigma-Aldrich) and

administered into the peritoneal cavity (2.0 mg per mouse) of the indicated mice at the indicated time point. All mice were bred in a temperature- and humidity-controlled specific pathogen-free animal facility at KAIST, with a 12-h light/dark cycle, and fed with *ad libitum* access to standard diet (PMI Lab diet) and water. All mice were anesthetized by intramuscular injection of a combination of anesthetics (80 mg/kg of ketamine and 12 mg/kg of xylazine) before being sacrificed.

### Histological analysis

On the indicated days after the treatments, mice were anesthetized by an intramuscular injection of combined anesthetics. Tissues including thyroid glands, trachea, and esophagus were fixed in 1% paraformaldehyde, embedded with tissue freezing medium (Leica) or paraffin, and sectioned. Samples were blocked with 5% goat (or donkey) serum in PBST (0.1% Triton X-100 in PBS) and incubated for 3 hr at room temperature (RT) with the following primary antibodies: anti-CD31 (hamster, clone 2H8, Millipore), anti-NG2 (rabbit polyclonal, Millipore), anti-GFP (rabbit polyclonal, Millipore), anti-VEGFR2 (rabbit, clone TO14, gifted by Dr. Rolf A. Brekken), anti-VEGFR3 (goat polyclonal, R&D Systems), anti-Ang2 (human monoclonal, clone 4H10; Han *et al*, 2016), anti-PH3 (rabbit polyclonal, Millipore), anti-caspase-3 (rabbit polyclonal, R&D Systems), and anti-GLUT1 (rabbit polyclonal, Millipore). After several washes with PBS, sections were incubated for 2 h at room temperature with one or more of the following secondary antibodies: Cy3- or FITC-conjugated anti-hamster IgG (Jackson ImmunoResearch), Cy3- or FITC-conjugated anti-rabbit IgG (Jackson ImmunoResearch), Cy3-conjugated anti-rat IgG (Jackson ImmunoResearch), and Cy3- or FITC-conjugated anti-goat IgG (Jackson ImmunoResearch). Nuclei were stained with 4′,6-diamidino-2-phenylindole (DAPI; Invitrogen). Zeiss LSM 510 confocal microscope equipped with argon and helium–neon lasers (Carl Zeiss) was used to visualize the fluorescent signals and to obtain digital images. Three-dimensional image reconstruction of the confocal-acquired *z*-stack images was performed using Imaris software (Bitplane AG, Zurich, Switzerland). This software allowed the reconstruction of an isosurface for each fluorescent channel. To examine β-galactosidase activity, the cryosections were incubated with a staining solution [2 mM magnesium chloride, 5 mM potassium ferricyanide, 5 mM potassium ferrocyanide, and 1 mg/ml 4-chloro-5-bromo-3-indolyl-β-D-galactopyranoside (X-gal) in PBS] at 37°C for 1–12 h. Paraffin-sectioned tissues were stained according to standard hematoxylin and eosin (H&E) staining protocol. The H&E images were captured by a microscope equipped with CCD camera (Carl Zeiss). To detect hypoxic areas in thyroid glands, Hypoxyprobe-1™ (60 mg/kg, solid pimonidazole hydrochloride, Natural Pharmacia International) was intraperitoneally injected 60 min before analyses. The thyroid glands were then harvested, sectioned, and stained with FITC-conjugated anti-Hypoxyprobe antibody.

### Preparations and treatments of reagents

The modulation of TSHR signaling in thyroid glands was achieved by modulating the pituitary secretion of its ligand, TSH, through the administration of T4 (Sigma) or PTU (Sigma) as previously

described (Franco *et al*, 2011). In detail, to suppress the pituitary secretion of TSH, T4 was dissolved in 4 M ammonium hydroxide and administered to the mice via drinking water at an estimated dose of 0.6 mg/kg/day. To increase the pituitary secretion of TSH, PTU was dissolved in 2.8 M ammonium hydroxide and administered to the mice via drinking water at an estimated dose of 0.72 g/kg/day. All tested mice were sacrificed and examined at indicated days. To produce recombinant proteins dimeric-Fc (Fc) and VEGF-Trap, stable CHO cell lines that secrete these recombinant proteins were used as previously described (Kim *et al*, 2013). A hybridoma cell line that expresses rat anti-VEGFR2 Ab, DC101, was purchased from American Type Culture Collection (ATCC). Hybridoma cells were grown in serum-free medium. Recombinant proteins in the supernatants were purified by column chromatography with Protein A agarose gel (Oncogene) using acid elution. After purification, the recombinant proteins were quantified using the Bradford assay and confirmed by Coomassie blue staining after SDS–PAGE. Proteins were subcutaneously injected with the following dosages and schedules: VEGF-Trap (25 mg/kg), DC101 (25 mg/kg), and Fc (25 mg/kg) every 2 days for 2 weeks.

### Measurement of serum TSH

Peripheral blood was collected via tail vein by incision at the indicated times. Blood was clotted for 30 min and centrifuged for 15 min at 3,300 *g*. Supernatant was collected and considered as serum. The collected samples were stored at −80°C before analysis. Serum TSH levels were measured by radioimmunoassay as previously described (Ying *et al*, 2005).

### Intravital imaging

Intravital imaging for thyroid gland vasculature was performed with the confocal imaging platform using a homebuilt microscope as previously described (Kim *et al*, 2008). Briefly, mice were anesthetized by intramuscular injection of combined anesthetics and surgical procedures were performed to expose thyroid glands followed by intravital imaging. To visualize blood vessels (BVs), 100 µl of FITC-conjugated dextran (5 mg/ml, 200 MDa, Sigma) was injected intravenously. To evaluate the blood flow in the thyroid glands, fluorescence-labeled RBCs using Vybrant® DiD Cell-Labeling Solution (Invitrogen) was adoptively transferred 10 min before intravital imaging. The confocal system involves two continuous-wave laser emitting at 488 nm (OBIS, Coherent) and 640 nm (Cube, Coherent) as excitation sources. The field of view (FOV) was 250 × 250 µm at the focal plane of a 40× objective lens (LUCPlanFL, numerical aperture (NA) = 0.6, Olympus). Images were displayed as 512 × 512 pixels per frame at a frame rate of 30 Hz and stored in a hard disk in real time by using a custom-written software and Matrox Imaging Library (MIL9).

### Transmission electron microscopy

Tissues of the thyroid glands were fixed with 2.5% glutaraldehyde (Ted Pella) in PBS overnight and washed five times with cacodylate buffer (0.1 M) containing 0.1% $CaCl_2$. Tissues were post-fixed for 2 h with 1% $OsO_4$ in 0.1 M cacodylate buffer (pH 7.2) and washed in cold distilled water. Tissue dehydration was conducted with ethanol series and propylene oxide, and the tissues were embedded in Embed-812 (EMS). Following resin polymerization for 36 h, the tissues were serially sectioned using an ULTRACUT UC7 ultramicrotome (Leica, Austria) and mounted on Formvar-coated slot grid. After staining with 4% uranyl acetate and lead citrate, the sections were observed using a Tecnai G2 Spirit Twin transmission electron microscope (FEI).

### Morphometric analysis

Morphometric analyses of thyroid glands were performed using ImageJ software (http://rsb.info.nih.gov/ij) or LSM Image Browser (Carl Zeiss). Heights of thyrocytes were averaged among 10 consecutive thyrocytes in H&E-stained images. For vascular densities, $CD31^+$ area was measured in 0.2-$mm^2$ areas and presented as a percentage of the total measured area. Vascular diameters were averaged among 10 consecutive FITC–dextran-perfused blood vessels in three random 0.05-$mm^2$ areas. Numbers of $PH3^+/CD31^+$ ECs or caspase-$3^+/CD31^+$ ECs were measured in three random 0.05-$mm^2$ areas. Numbers of fenestrae and caveolae in 10 µm vessel perimeter of the capillary ECs were measured on 5–8 random areas of TEM images (Kamba *et al*, 2006). Coverage of $NG2^+$ pericytes to BVs was calculated as $NG2^+$ area divided by $CD31^+$ BV area in three random 0.05-$mm^2$ areas of each thyroid gland and presented as a percentage. Expression of VEGFR2 and VEGFR3 was determined by measuring the signal densities in three random 0.053-$mm^2$ areas and presented as a relative percentage to the signal densities of CD31.

### Reagents, cells, and culture condition for *in vitro* studies

FRTL-5 cell line was purchased from American Type Culture Collection (ATCC). The cells were grown in 6H medium consisting of Coon's modified F-12 (Sigma) supplemented with 5% fetal bovine serum (Gibco), 1 mM nonessential amino acids (Gibco), and a mixture of six hormones as follows: bovine TSH (1 mU/ml, Sigma), bovine insulin (10 µg/ml, Sigma), hydrocortisone (0.4 ng/ml, Calbiochem), human transferrin (5 µg/ml, Sigma), glycyl-L-histidyl-L-lysine acetate (10 ng/ml, Sigma), and somatostatin (10 ng/ml, Sigma). Media were changed every 2–3 days, and cells were passaged every 7–10 days. In each experiment, the cells were incubated in 4H medium (which is devoid of insulin and TSH) with 0.2% FBS for 2 days and were treated with indicated doses of TSH for 3 h. To block each specific signaling pathway, the cells were pretreated with 3 µM KT5720 (protein kinase A inhibitor, Calbiochem), 0.1 µM Go6976 (protein kinase C inhibitor, Calbiochem), or 20 µM LY294002 (PI3K inhibitor, Calbiochem) in 0.1% DMSO-containing media 1 h prior to adding TSH (1 mU/ml). The cells cultured in 0.1% DMSO media without additional inhibitors were used as a control.

### Patients' data

Experimental procedures with human pathologic specimens were approved by the Institutional Review Board of Pusan National

**The paper explained**

**Problem**

Thyroid gland is a highly vascularized organ, in which unique organ-specific endothelial cell phenotypes are present including endothelial fenestrae and dynamic vascular remodeling. However, to date, any attempt that specifically targets thyroid vasculatures in pathology has yet to be conducted. Thus, it is reasonable to find out the signaling pathway that critically governs thyroid vascular phenotypes as it might provide a potential therapeutic target in thyroid disease.

**Results**

Among two major vascular ligands–receptor systems, VEGF–VEGFR2 was profoundly and distinctly expressed, while Ang–Tie2 was not significantly expressed in thyroid glands. Using genetically modified mice, we uncovered the indispensable roles of paracrine VEGF-A–VEGFR2 pathway governing the integrity of thyroid angiofollicular unit, while the role of VEGFR3 is dispensable. Importantly, VEGFR2 blockade not only attenuates the thyrotropin-dependent vascular remodeling, but also reduces thyrotropin responsiveness of follicular cells, which ultimately ameliorates thyroid goitrogenesis.

**Impact**

Microvessel densities and their VEGFR2 expression were strikingly increased in conditions of TSHR hyper-activation in mice and human. This study demonstrated that VEGFR2 inhibition through either genetic deletion or antibody-mediated blockade sufficiently reduces transcellular transport capacities of ECs and thyrotropin responsiveness of follicular cells leading to decreased thyroid weights in preclinical models of thyroid goitrogenesis. Our findings identify VEGFR2 blockade as a novel therapeutic avenue for targeting vasculatures in TSHR-associated thyroid disease.

University Hospital (PNUHIRB-017). All human samples were collected by the tissue bank of Pusan National University Hospital, Busan, Korea, with the informed consents from the donors following the bioethics and safety regulations. The diagnoses of the specimens included Graves' diseases ($n = 4$) and papillary thyroid cancer ($n = 5$) for evaluation of normal contralateral lobe of the thyroid gland. The formalin-fixed, paraffin-embedded sectioned tissues were used to examine BV density and VEGFR2 expression. Immunostaining was carried out using the Bond-Max autostainer and reagents (Vision Biosystems). Deparaffinization was done automatically in the autostainer with Bond Wash solutions for 30 min. The slides were then incubated with Epitope Retrieval Solution 1 (Leica Biosystems, Wetzlar, Germany) for 20 min, peroxide block for 5 min, primary monoclonal antibody for 15 min, post-primary reagent for 8 min, and polymer for 8 min. Anti-VEGFR2 (55B11, Cell Signaling Technology) and anti-CD31 (JC70A, DAKO) were used as the primary antibodies. To measure number of CD31[+] microvessels in thyroid glands, three areas with the highest concentration of BVs were selected after scanning at low-power field (40×). Then, each area was evaluated with high-power field (400×) and the average number of microvessels per high-power field was determined. VEGFR2 immunostaining score was measured by a pathologist according to a semi-quantitative manner as a sum of the intensity (score, 0–3) and proportion (score, 0–3) of immunoreactions. Proportion was scored for the percentage of CD31[+] BVs (0, 0; 1, < 5%; 2, 5–50%; 3, > 50%).

## Statistical analysis

Sample sizes were estimated by a power analysis ("TrialSize" package of R3.2.3 version) to detect a 20–40% difference ($\delta$) with a significance level ($\alpha$) of 0.05 and power ($\beta$) of 0.8. Animals or samples were not randomized during experiments and not excluded from analyses. The investigators were not blinded to group allocation during experiments and outcome analyses. Values are presented as mean ± standard deviation (s.d.). Significant differences between the groups were determined by two-sided Mann–Whitney *U*-test between two groups. In case of multiple groups, significant differences between the groups were determined by Kruskal–Wallis test with Tukey's multiple comparison test. Statistical analysis was performed using PASW Statistics 18 (SPSS). Statistical significance was set at $P < 0.05$.

**Expanded View** for this article is available online.

## Acknowledgements

This study was supported by the Institute of Basic Science funded by the Ministry of Science, ICT and Future Planning (MSIP), Korea (IBS-R025-D1-2016, G.Y.K.).

## Author contributions

JYJ, SYC, IP, DYP, KC, YKK, and J-WP designed and performed the experiments and analyzed the data; PK, B-JL, S-YC, YJP, and MS designed the experiments and provided critical comments on this study; MH, YK, AN, and KA provided the mice and critical comments on this study; JYJ, SYC, and GYK generated the figures and wrote and edited the manuscript; and MS and GYK directed and supervised the project.

## Conflict of interest

The authors declare that they have no conflict of interest.

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
