## [Review Process File · EMBO Molecular Medicine]

VEGFR2 but not VEGFR3 governs integrity and remodeling of thyroid angiofollicular unit in normal state and during goitrogenesis

Jeon Yeob Jang, Sung Yong Choi, Intae Park, Do Young Park, Kibaek Choe, Pilhan Kim, Young Keum Kim, Byung-Joo Lee, Masanori Hirashima, Yoshiaki Kubota, Jeong-Won Park, Sheue-Yann Cheng, Andras Nagy, Young Joo Park, Kari Alitalo, Minho Shong & Gou Young Koh

Corresponding author: Gou Young Koh, Institute of Basic Science

Review timeline:

Submission date:	13 November 2016
Editorial Decision:	20 December 2016
Revision received:	03 March 2017
Editorial Decision:	15 March 2017
Revision received:	19 March 2017
Accepted:	21 March 2017

Transaction Report:

Editor: Roberto Buccione

1st Editorial Decision

20 December 2016

Thank you for the submission of your manuscript to EMBO Molecular Medicine. We have now heard back from the three Reviewers whom we asked to evaluate your manuscript.

You will see that the Reviewers are quite supportive of your work and list a limited number of points for your consideration and action. I will not dwell into any detail, as the evaluations are self-explanatory. Please note that reviewer 3 would like you provide further experimental data concerning the analysis of tissue hypoxia and HIF1 α localisation.

Considered all the above, while publication of the paper cannot be considered at this stage, we would be pleased to consider a revised submission, with the understanding that the Reviewers' concerns must be addressed as outlined above, with additional experimental data where appropriate and that acceptance of the manuscript will entail a second round of review.

In the likely case that your manuscript will be accepted for publication, we would be requiring a number of editorial amendments to adhere to our author guidelines. I suggest that you do so already in your next version to accelerate the editorial process. Please therefore, take action on the following:

- 1) Figures should be provided as individual files (1 file per figure).

- 2) Movies should be labeled as Movie EV1, Movie EV2, etc. and be ZIPped together with their legends. The legends should be removed from the appendix file (<http://embomolmed.embopress.org/authorguide#expandedview>). Please correct the movie callouts in the main text accordingly.
- 3) As per our Author Guidelines, the description of all reported data that includes statistical testing must state the name of the statistical test used to generate error bars and P values, the number (n) of independent experiments underlying each data point (not replicate measures of one sample), and the actual P value for each test (not merely 'significant' or 'P < 0.05').
- 4) The manuscript must include a statement in the Materials and Methods identifying the institutional and/or licensing committee approving the experiments, including any relevant details (like how many animals were used, of which gender, at what age, which strains, if genetically modified, on which background, housing details, etc). We encourage authors to follow the ARRIVE guidelines for reporting studies involving animals. Please see the EQUATOR website for details: <http://www.equator-network.org/reporting-guidelines/improving-bioscience-research-reporting-the-arrive-guidelines-for-reporting-animal-research/>. Please make sure that ALL the above details are reported.
- 5) We encourage the publication of source data, with the aim of making primary data more accessible and transparent to the reader. Would you be willing to provide a PDF file per figure that contains the original, uncropped and unprocessed scans of all or at least the key gels used in the manuscript and/or source data sets for relevant graphs? The files should be labeled with the appropriate figure/panel number, and in the case of gels, should have molecular weight markers; further annotation may be useful but is not essential. The files will be published online with the article as supplementary "Source Data" files. If you have any questions regarding this just contact me.
- 6) Every published paper includes a 'Synopsis' to further enhance discoverability. Synopses are displayed on the journal webpage and are freely accessible to all readers. They include a short standfirst as well as 2-5 one sentence bullet points that summarise the paper. Please provide the synopsis including the short list of bullet points that summarise the key NEW findings. The bullet points should be designed to be complementary to the abstract - i.e. not repeat the same text. We encourage inclusion of key acronyms and quantitative information. Please use the passive voice. Please attach this information in a separate file or send them by email, we will incorporate it accordingly. You are also welcome to suggest a striking image or visual abstract to illustrate your article. If you do please provide a jpeg file 550 px-wide x 400-px high.
- 7) We now mandate that ALL corresponding authors list an ORCID digital identifier. You may acquire one through our web platform upon submission and the procedure takes <90 seconds to complete. We also encourage co-authors to supply an ORCID identifier, which will be linked to their name for unambiguous name identification.

I look forward to seeing a revised form of your manuscript as soon as possible.

***** Reviewer's comments *****

Referee #1 (Comments on Novelty/Model System):

The submitted manuscript is of high interest as deciphering a new molecular mechanism regulating the coordinated interaction between TSH, thyroid epithelial cell function and thyroid vasculature in normal state and disease.
The study was performed using several relevant conditional knock-out models providing convincing evidence for their conclusions.

Referee #1 (Remarks):

Jang et al present a fundamental study on the role of VEGF-A and VEGFR2 and 3 for homeostasis of the angiofollicular unit of the thyroid gland. They used different transgenic mouse models to

investigate the interaction of TSH and thyroid capillaries during hypo- and hyperthyroidism. They provide a detailed set of experiments deciphering the TSH dependent molecular mechanisms regulating functional adaptation of the thyroid angiofollicular unit to euthyroidism, hypo- or hyperthyroidism.

The article is of great interest for a better understanding of vascular regulation of thyroid function in normal state and disease. It further provides important insights how anti-angiogenic substances cause hypothyroidism and how such medications could be used for treatment of Graves' diseases. In summary, the manuscript adds to current knowledge a new molecular mechanism regulating thyroid function.

The article is well written. The methodological part is detailed and precise. The results are presented clearly in text and figure. Figure legends are precise. Statistical analyses are accurate. The claims of the authors are clearly supported by the experimental data. The discussion is concise and discusses the majority of the relevant literature.

Minor comments:

1. Abstract: p3 line 7: please replace "grand" by "gland".
2. Figure 1: Please correct discrepancy in Figure legend (p30) and Figure 1E concerning abbreviations (figure legend Tr=trachea, Fig 1E T=trachea, figure legend Es=esophagus, Fig 1E E=esophagus).

Referee #2 (Remarks):

This paper describes a comprehensive series of experiments aimed at deciphering the molecular mechanisms modulating thyroid vasculature, and specifically endothelial fenestrae, which are a characteristic of this vasculature. The data show that VEGF-A and VEGFR2 are expressed in follicular cells and vascular cells, respectively, and are stimulated by serum TSH, the levels of which were manipulated by feeding mice PTU and T4. Blocking VEGF-A or VEGFR2 by pharmacological, immunological or genetic approaches mitigated TSH-induced goitrogenesis. Lastly, in the thyroid of 4 patients with Graves disease, a model of TSHR hyperstimulation, VEGFR2 was overexpressed compared to normal tissue (the unaffected lobe of 4 patients thyroidecomized for papillary cancer). Overall, the amount of data is impressive and the conclusions are drawn logically.

Specific comment: in Figure 7 F-M, specify that the panels labelled as 'control' are control mice, not mice untreated with PTU.

Referee #3 (Comments on Novelty/Model System):

It is a high quality study that clearly shows that continuous production of Vegf-a by follicular cells is necessary for maintenance of vasculature and the entire thyroid function.

Referee #3 (Remarks):

The manuscript of Jang et al. investigates the cross-talk of thyroid blood vessels and thyroid follicular epithelial cells using multiple approaches, such as conditional inactivation of Vegfr-2 or Vegfr-3 in endothelial cells, VEGF-trap of Vegfr-2 blocking antibodies. In sum, they show that continuous activation of Vegfr2 but not Vegfr-3 is important for maintenance of thyroid vasculature, and loss of thyroid vessels results in reduction of thyroid weight. On the other hand, blocking vascular expansion prevents expansion of follicular cells in response to PTU. They also show that blood vascular density and Vegfr-2 staining are increased in Grave's disease, suggesting that this mechanism is relevant for human pathology.

Overall, this is a high quality work, which provides a detailed description of the thyroid vasculature and its role in maintenance of thyroid function, which significantly expands the initial observations by e.g. Kamba et al. on the sensitivity of thyroid vasculature to anti-VEGF agents. Another valuable observation is the demonstration that although Vegfr-3 is also highly expressed in thyroid vessels, it

has little, if any role, in the regulation vessel expansion, unlike it was observed in mouse retina, or in formation of endothelial fenestrae.

My comments are outlined below:

1. The authors show that there is a direct induction of VEGF-A expression by TSH, at least in culture. In many other systems VEGF-A expression is induced in response to tissue hypoxia and stabilization of HIF-1alpha. It may be important to make this distinction here and add the analysis of tissue hypoxia and HIF1a localization in thyroids following T4- or PTU treatment.
2. Vegf-A is known to induce the expression of angiopoietin-2 in endothelial cells both in vitro and in vivo, therefore a virtual absence of Angpt2 expression in thyroid angiogenic vasculature is surprising - what is the reason for this? Could the authors discuss it in more detail?
3. Previous studies demonstrated that deletion of Vegfr-2 in blood endothelial cells induces Vegfr-3 protein (Benedito et al., 2012) and the genetic inactivation of Vegfr-3 in retinal blood endothelium results was shown to induce hypersprouting (Nat Cell Biol paper of Alitalo's group) - whereas these parameters do not seem to change in thyroid vasculature of Vegfr2ecKO or Vegfr3ecKO mice? This is an important point to discuss in order to present current findings in the context of existing knowledge.

Minor comments

1. Page 10 VEGF-A trap is described as a selective VEGF-A blocking reagent, however it will also bind Plgf and Vegf-B - please correct this.
2. Page 11 "Furthermore, increments of the thyroid weights after PTU administration ..." - please include a reference to the figure with these data.
3. Page 11 - did the authors really use Fc as a control to DC101 treatment? Why not same isotype/species IgG (rat in this case)?

1st Revision - authors' response

03 March 2017

Answers to the editor's and reviewers' comments (EMM-2016-07341R1)

We deeply appreciate the editor and reviewers for their thoughtful, critical and constructive comments, which have undoubtedly provided us with valuable opportunities to improve our work. We have performed additional experiments and revised the manuscript to address the issues raised by the reviewers.

Referee #1 (Comments on Novelty/Model System): The submitted manuscript is of high interest as deciphering a new molecular mechanism regulating the coordinated interaction between TSH, thyroid epithelial cell function and thyroid vasculature in normal state and disease. The study was performed using several relevant conditional knock-out models providing convincing evidence for their conclusions.

Referee #1 (Remarks):Jang et al present a fundamental study on the role of VEGF-A and VEGFR2 and 3 for homeostasis of the angiofollicular unit of the thyroid gland. They used different transgenic mouse models to investigate the interaction of TSH and thyroid capillaries during hypo- and hyperthyroidism. They provide a detailed set of experiments deciphering the TSH dependent molecular mechanisms regulating functional adaptation of the thyroid angiofollicular unit to euthyroidism, hypo- or hyperthyroidism.

The article is of great interest for a better understanding of vascular regulation of thyroid function in normal state and disease. It further provides important insights how anti-angiogenic substances cause hypothyroidism and how such medications could be used for treatment of Graves' diseases. In summary, the manuscript adds to current knowledge a new molecular mechanism regulating thyroid function.

The article is well written. The methodological part is detailed and precise. The results are presented clearly in text and figure. Figure legends are precise. Statistical analyses are accurate. The claims of the authors are clearly supported by the experimental data. The discussion is concise and discusses the majority of the relevant literature.

We appreciate these encouraging and thoughtful comments.

Minor comments:

1. Abstract: p3 line 7: please replace "grand" by "gland".

We corrected this error in the revised manuscript.

2. Figure 1: Please correct discrepancy in Figure legend (p30) and Figure 1E concerning abbreviations (figure legend Tr=trachea, Fig 1E T=trachea, figure legend Es=esophagus, Fig 1E E=esophagus.

We corrected the abbreviations in the Figure 1E of the revised Figures.

Referee #2 (Remarks):

This paper describes a comprehensive series of experiments aimed at deciphering the molecular mechanisms modulating thyroid vasculature, and specifically endothelial fenestrae, which are a characteristic of this vasculature. The data show that VEGF-A and VEGFR2 are expressed in follicular cells and vascular cells, respectively, and are stimulated by serum TSH, the levels of which were manipulated by feeding mice PTU and T4. Blocking VEGF-A or VEGFR2 by pharmacological, immunological or genetic approaches mitigated TSH-induced goitrogenesis. Lastly, in the thyroid of 4 patients with Graves disease, a model of TSHR hyperstimulation, VEGFR2 was overexpressed compared to normal tissue (the unaffected lobe of 4 patients thyroidecomized for papillary cancer). Overall, the amount of data is impressive and the conclusions are drawn logically.

We appreciate these encouraging and thoughtful comments.

Specific comment: in Figure 7 F-M, specify that the panels labelled as 'control' are control mice, not mice untreated with PTU.

We appreciate this constructive comment. We correct this error by specifying the wild type and *VEGFR2*^{ΔEC} mice administered with PTU in the revised Figure 7.

Referee #3 (Remarks):

The manuscript of Jang et al. investigates the cross-talk of thyroid blood vessels and thyroid follicular epithelial cells using multiple approaches, such as conditional inactivation of Vegfr-2 or Vegfr-3 in endothelial cells, VEGF-trap or Vegfr-2 blocking antibodies. In sum, they show that continuous activation of Vegfr2 but not Vegfr-3 is important for maintenance of thyroid vasculature, and loss of thyroid vessels results in reduction of thyroid weight. On the other hand, blocking vascular expansion prevents expansion of follicular cells in response to PTU. They also show that blood vascular density and Vegfr-2 staining are increased in Grave's disease, suggesting that this mechanism is relevant for human pathology.

Overall, this is a high quality work, which provides a detailed description of the thyroid vasculature and its role in maintenance of thyroid function, which significantly expands the initial observations by e.g. Kamba et al. on the sensitivity of thyroid vasculature to anti-VEGF agents. Another valuable observation is the demonstration that although Vegfr-3 is also highly expressed in thyroid vessels, it has little, if any role, in the regulation vessel expansion, unlike it was observed in mouse retina, or in formation of endothelial fenestrae.

We appreciate these encouraging and thoughtful comments.

My comments are outlined below:

1. The authors show that there is a direct induction of VEGF-A expression by TSH, at least in culture. In many other systems VEGF-A expression is induced in response to tissue hypoxia and stabilization of HIF-1α. It may be important to make this distinction here and add the analysis of tissue hypoxia and HIF1α localization in thyroids following T4- or PTU treatment.

Thank you for this insightful comment. We performed additional experiments to measure tissue hypoxia using pimonidazole (Hypoxyprobe) and also analyzed glucose transporter 1 (GLUT1, alternative hypoxia marker) expression. However, almost no Hypoxyprobe or GLUT1 were detected in the thyroid glands treated with T4 or PTU, while they were highly detected in the hypoxic regions of implanted glioblastoma (positive control). We included these results (Appendix Figures S4) and their descriptions (page 9) into the revised manuscript.

In addition, we further evaluated tissue hypoxic status within the thyroid gland because VEGF-A expression has been known to be induced in response to tissue hypoxia in many other systems (Ferrara, 2004). However, tissue hypoxia, as quantified using Hypoxyprobe-1 or by GLUT1 expression (an alternative hypoxia marker), was not meaningfully observed within the thyroid gland of control, T4-treated, or PTU-treated mice, while they were highly detected in the hypoxic regions of implanted glioblastoma (positive control) (Appendix Fig S4).

2. Vegf-A is known to induce the expression of angiopoietin-2 in endothelial cells both in vitro and in vivo, therefore a virtual absence of Angpt2 expression in thyroid angiogenic vasculature is surprising - what is the reason for this? Could the authors discuss it in more detail?

We agree with this critical comment. The virtual absence of Ang2 expression in endothelial cells despite the robust expression of VEGF-VEGFR2 in thyroid gland is another intriguing aspect of this study. Our additional analyses showed that endothelial Ang2 expression was increased in response to PTU administration while it was not observed in normal thyroid glands (Figure for reviewer). Based on these results, we could speculate that the pathologic increase in VEGF-VEGFR2 signaling directly and indirectly enhances endothelial Ang2 expression in the thyroid gland, while physiological level of VEGF-VEGFR2 is insufficient to induce Ang2 expression. We also performed Tie2 immunostaining which indicated that Tie2 was not meaningfully detected or changed in thyroid glands of control, T4-treated, and PTU-treated mice. Therefore, it is puzzling as to what the roles of the increased Ang2 could be without Tie2 in the thyroid glands of PTU-treated mouse, which can be evaluated in the future. We are showing this data only for the reviewer because it is somewhat beyond the scope of this study (VEGF-VEGFR system).

Endothelial expression of Ang2 is increased in PTU-induced thyroid goiter.

A-D: Images and comparisons of Ang2 and Tie2 expression in thyroid glands of control (Co), T4-treated (T4), or PTU-treated (PTU) mice. Scale bars, 50 μ m. Error bars represent mean \pm s.d. Each group, n = 4. * p < 0.05 versus control by Kruskal-Wallis test.

3. Previous studies demonstrated that deletion of Vegfr-2 in blood endothelial cells induces Vegfr-3 protein (Benedito et al., 2012) and the genetic inactivation of Vegfr-3 in retinal blood endothelium results was shown to induce hypersprouting (Nat Cell Biol paper of Alitalo's group) - whereas these parameters do not seem to change in thyroid vasculature of Vegfr2ecKO or Vegfr3ecKO mice? This is an important point to discuss in order to present current findings in the context of existing knowledge.

Following the reviewer's suggestion, we reviewed the suggested paper (Benedito et al., 2012). However, we found that deletion of Vegfr2 in blood endothelial cells also decreased Vegfr3 expression in Fig 2 of the paper, which is consistent with our data (Fig 4D and E).

We also reviewed the other suggested paper (Nat Cell Biol paper of Alitalo's group) showing that the genetic inactivation of Vegfr3 in retinal blood endothelium induces hypersprouting which is discrepant with our data. However, the study evaluated the changes in growing retinal vasculatures during postnatal development, while our study evaluated the changes in the quiescent thyroid vasculatures during adulthood. Moreover, we suggest that different organ-specific characteristics of ECs may influence the phenotype in Vegfr3^{ecKO} mice. We included these explanation in the revised discussion (Page 16)

These results appear to be discrepant with a previous observation that genetic depletion of endothelial *VEGFR3* induces hypersprouting of growing retinal vessels (Tammela et al, 2011). While the previous study evaluated the changes in growing retinal vessels during postnatal development, our model system involved quiescent thyroid vasculatures during adulthood. Thus, VEGFR3 may play an indispensable role in thyroid vasculature during development period, which requires further evaluation.

Minor comments

1. Page 10 VEGF-A trap is described as a selective VEGF-A blocking reagent, however it will also bind Plgf and Vegf-B - please correct this.

We appreciate this constructive comment. We included an additional description to address the issue in the revised results (Page 10).

We further evaluated the role of VEGF-A in adult normal thyroid glands by using a VEGF-Trap (Holash et al, 2002), which has been known to block VEGF-A, VEGF-B, and PlGF (placental growth factor).

2. Page 11 "Furthermore, increments of the thyroid weights after PTU administration ... " - please include a reference to the figure with these data.

We included additional data for reference (Appendix Fig S6G and H).

3. Page 11 - did the authors really use Fc as a control to DC101 treatment? Why not same isotype/species IgG (rat in this case)?

Thank you for pointing out this issue. We overlooked this caveat. As we described in the Method (page 19), dimeric-Fc (Fc portion of human IgG) produced from stable CHO cell line was used as a control protein for VEGF-Trap and DC101. In addition, our pilot experiment revealed no apparent differences between human dimeric-Fc and rat dimeric-Fc (IgG), at least in thyroid vasculatures.

Reference

Ferrara N (2004) Vascular endothelial growth factor: basic science and clinical progress. *Endocr Rev* 25: 581-611

Holash J, Davis S, Papadopoulos N, Croll SD, Ho L, Russell M, Boland P, Leidich R, Hylton D, Burova E et al (2002) VEGF-Trap: a VEGF blocker with potent antitumor effects. *Proceedings of the National Academy of Sciences of the United States of America* 99: 11393-11398

Tammela T, Zarkada G, Nurmi H, Jakobsson L, Heinolainen K, Tvorogov D, Zheng W, Franco CA, Murtomaki A, Aranda E et al (2011) VEGFR-3 controls tip to stalk conversion at vessel fusion sites by reinforcing Notch signalling. *Nature cell biology* 13: 1202-1213

2nd Editorial Decision

15 March 2017

Thank you for the submission of your revised manuscript to EMBO Molecular Medicine. We have now received the enclosed report from the reviewer who was asked to re-assess it. As you will see the reviewers are now globally supportive and I am pleased to inform you that we will be able to accept your manuscript pending the following final amendments:

1. Please provide all figures in portrait orientation. Some are currently in landscape format.
2. Please make sure that cited references with multiple authors list 20 names (not 10 et al.)
3. We note that the quality of some images especially of the blots is not ideal, this is especially the case for figure 1. The resolution appears low and the bands appear blocky/blurry when magnifying. Please use better images as these issues could lead to problems when the production team tries to resize these images for the final manuscript.
4. Please update the manuscript callouts for movies EV1 to EV3 as they are currently indicated as Appendix movies.
5. Please add a scale bar to the Thyroid panel in figure S1C
6. As per our Author Guidelines, the description of all reported data that includes statistical testing must state the name of the statistical test used to generate error bars and P values, the number (n) of independent experiments underlying each data point (not replicate measures of one sample), and the actual P value for each test (not merely 'significant' or ' $P < 0.05$ ').
7. The manuscript must include a statement in the Materials and Methods identifying the institutional and/or licensing committee approving the experiments, including any relevant details (like how many animals were used, of which gender, at what age, which strains, if genetically modified, on which background, housing details, etc). We encourage authors to follow the ARRIVE guidelines for reporting studies involving animals. Please see the EQUATOR website for details: <http://www.equator-network.org/reporting-guidelines/improving-bioscience-research-reporting-the-arrive-guidelines-for-reporting-animal-research/>. Please make sure that ALL the above details are reported.
8. We encourage the publication of source data, with the aim of making primary data more accessible and transparent to the reader. Would you be willing to provide a PDF file per figure that contains the original, uncropped and unprocessed scans of all or at least the key gels used in the manuscript and/or source data sets for relevant graphs? The files should be labeled with the appropriate figure/panel number, and in the case of gels, should have molecular weight markers; further annotation may be useful but is not essential. The files will be published online with the article as supplementary

Please submit your revised manuscript within two weeks. I look forward to seeing a revised form of your manuscript as soon as possible.

***** Reviewer's comments *****

Referee #3 (Comments on Novelty/Model System):

No ethical concerns

Referee #3 (Remarks):

The authors responded well to my remarks and improved the manuscript, I have no further comments

2nd Revision - authors' response

19 March 2017

(See next page)

Point-by-point responses:

1. Please provide all figures in portrait orientation. Some are currently in landscape format.

All figures have been changed to portrait format.

2. Please make sure that cited references with multiple authors list 20 names (not 10 et al.)

The references have been edited accordingly.

3. We note that the quality of some images especially of the blots is not ideal, this is especially the case for figure 1. The resolution appears low and the bands appear blocky/blurry when magnifying. Please use better images as these issues could lead to problems when the production team tries to resize these images for the final manuscript.

Thank you for notifying us with this problem. We have identified the cause of this problem and have updated our figures with better resolution images.

4. Please update the manuscript callouts for movies EV1 to EV3 as they are currently indicated as Appendix movies.

Manuscript callouts for movies have been edited in the main text.

5. Please add a scale bar to the Thyroid panel in figure S1C

The scale bar has been added to figure S1C.

6. As per our Author Guidelines, the description of all reported data that includes statistical testing must state the name of the statistical test used to generate error bars and P values, the number (n) of independent experiments underlying each data point (not replicate measures of one sample), and the actual P value for each test (not merely 'significant' or ' $P < 0.05$ ').

We edited and added these information where appropriate in our updated figures and supplementary figures as well as their respective legends.

7. The manuscript must include a statement in the Materials and Methods identifying the institutional and/or licensing committee approving the experiments, including any relevant details (like how many animals were used, of which gender, at what age, which strains, if genetically modified, on which background, housing details, etc). We encourage authors to follow the ARRIVE guidelines for reporting studies involving animals. Please see the EQUATOR website for details: <http://www.equator-network.org/reporting-guidelines/improving-bioscience-research-reporting-the-arrive-guidelines-for-reporting-animal-research/>. Please make sure that ALL the above details are reported.

All the relevant details, such as the approval of experiments and the animals used for the study, have been updated in the revised Materials and Methods section.

8. We encourage the publication of source data, with the aim of making primary data more

accessible and transparent to the reader. Would you be willing to provide a PDF file per figure that contains the original, uncropped and unprocessed scans of all or at least the key gels used in the manuscript and/or source data sets for relevant graphs? The files should be labeled with the appropriate figure/panel number, and in the case of gels, should have molecular weight markers; further annotation may be useful but is not essential. The files will be published online with the article as supplementary

We uploaded the source data for Figure 1B (unprocessed gel image) with appropriate labels and markings. If you require any other source data, please let us know as we are happy to provide all necessary information for the transparency of our manuscript.

Corresponding Author Name: Gou Young Koh, Minh Shong

Manuscript Number: EMM-2016-07341